# A Review of In Situ Quality Monitoring in Additive Manufacturing Using Acoustic Emission Technology

**DOI:** 10.3390/s26020438

**Published:** 2026-01-09

**Authors:** Wenbiao Chang, Qifei Zhang, Wei Chen, Yuan Gao, Bin Liu, Zhonghua Li, Changying Dang

**Affiliations:** 1School of Mechanical Engineering, North University of China, Taiyuan 030051, China; 2Shanxi Provincial Key Laboratory for Controlled Metal Solidification and Additive Manufacturing, North University of China, Taiyuan 030051, China; 3Chongqing Qingshan Industrial Co., Ltd., Chongqing 402776, China; 4School of Materials Science and Engineering, North University of China, Taiyuan 030051, China

**Keywords:** metal additive manufacturing, acoustic emission, in-process monitoring, defect detection

## Abstract

Additive manufacturing (AM) has emerged as a pivotal technology in component fabrication, renowned for its capabilities in freeform fabrication, material efficiency, and integrated design-to-manufacturing processes. As a critical branch of AM, metal additive manufacturing (MAM) has garnered significant attention for producing metal parts. However, process anomalies during MAM can pose safety risks, while internal defects in as-built parts detrimentally affect their service performance. These concerns underscore the necessity for robust in-process monitoring of both the MAM process and the quality of the resulting components. This review first delineates common MAM techniques and popular in-process monitoring methods. It then elaborates on the fundamental principles of acoustic emission (AE), including the configuration of AE systems and methods for extracting characteristic AE parameters. The core of the review synthesizes applications of AE technology in MAM, categorizing them into three key aspects: (1) hardware setup, which involves a comparative analysis of sensor selection, mounting strategies, and noise suppression techniques; (2) parametric characterization, which establishes correlations between AE features and process dynamics (e.g., process parameter deviations, spattering, melting/pool stability) as well as defect formation (e.g., porosity and cracking); and (3) intelligent monitoring, which focuses on the development of classification models and the integration of feedback control systems. By providing a systematic overview, this review aims to highlight the potential of AE as a powerful tool for real-time quality assurance in MAM.

## 1. Introduction

### 1.1. Metal Additive Manufacturing Technologies

Additive Manufacturing (AM) is a process that constructs three-dimensional objects layer-by-layer from a digital model, following a “bottom-up” approach contrary to traditional subtractive methods [1,2,3]. This technology, also known as “3D Printing”, “Rapid Prototyping”, or “Solid Freeform Fabrication” depending on specific process characteristics [4], has matured significantly with advancements in its core pillars: material design, structural geometry, computational tools and interfaces, and manufacturing processes [5]. Consequently, the spectrum of printable materials has expanded enormously, now encompassing metals, non-metals, composites, various functional materials, as well as biomaterials and even living tissues [6]. Simultaneously, the energy sources utilized for fabrication have diversified to include lasers, electron beams, specific wavelength sources, arcs, and their combinations [7]. These developments have propelled AM into widespread applications across diverse fields such as biomedical engineering, aerospace, and architectural conservation [8].

Metal additive manufacturing (MAM), an important branch in the field of AM, encompasses a variety of specific processing methods. Its classification is shown in Figure 1. According to different deposition methods, it is divided into two major categories: powder bed fusion (PBF) and directed energy deposition (DED). The PBF technology selectively melts the pre-prepared powder on the powder bed, excelling in manufacturing high-precision and complex-shaped end parts with excellent density and surface quality. However, its forming size is limited and the cost is relatively high. The DED technology directly feeds the powder or wire into the molten pool for deposition, featuring a high printing rate, the ability to repair large-sized parts, good flexibility, but with lower forming accuracy and requiring a large amount of subsequent processing. PBF techniques are further classified according to the energy source employed, namely laser beam powder bed fusion (LB-PBF) and electron beam powder bed fusion (EB-PBF). The LB-PBF process can be subdivided based on laser power and the extent of material melting into selective laser sintering (SLS) and selective laser melting (SLM). Conversely, DED techniques utilize a wider range of energy sources, including laser directed energy deposition (L-DED), electron beam directed energy deposition (EB-DED), and wire arc directed energy deposition (WA-DED).

In addition to the aforementioned AM methods, there are Binder Jetting Additive Manufacturing (BJ-AM) [9], Material Jetting Additive Manufacturing (MJ-AM) [10], and Material Extrusion Additive Manufacturing (MEX-AM) [11]. MEX-AM is widely popular due to its low cost and ease of use; MJ-AM can manufacture high-precision parts with smooth surfaces and multi-material properties through a precise curing process similar to inkjet printing; and BJ-AM uses binders to selectively bond powders, achieving unsupported, full-color, and efficient large-scale manufacturing. These technologies, from different aspects such as economy, precision, and efficiency, jointly support a wide range of industrial and prototyping manufacturing needs.

In the research of BJ-AM, Kumar et al. [12,13] improved the part density from 92% after sintering to 99.7% of the theoretical density through post-treatment with hot isostatic pressing (HIP), and further prepared copper parts with porosity ranging from 2.7% to 16.4% by adjusting the powder morphology, sintering process, and HIP parameters. They systematically revealed the influence laws of porosity on material properties. In MJ-AM, non-selective ultraviolet (UV) irradiation can easily lead to uneven dose received by components at different heights, causing fluctuations in mechanical properties and batch-to-batch differences. To control the process fluctuations, Bezek et al. [14] developed a process model that combines irradiance distribution and molding configuration. Based on the Bayesian-Lambert theorem, they quantitatively predicted the cumulative UV dose of each printed voxel and established the correlation between dose and mechanical properties through experiments. In the field of MEX-AM, Ren et al. [15] systematically studied the key performance parameters such as the rheological behavior of raw materials, the strength of green bodies, and the hardness of sintering through the melt extrusion printing technology, and optimized the printing and sintering parameters using the orthogonal design method. In terms of process mechanism research, Ajjarapu et al. [16] constructed a process diagram to reveal the important role of back pressure and reflux in the pressure drop of the nozzle, and by means of variance analysis and regression models, clarified the influence of printing conditions on performance and prediction ability.

As the earliest additive technology to be industrialized, SLS utilizes the liquid-phase solidification of low-melting-point metal powders to re-bond high-melting-point metal powder particles [17,18]. However, the formed parts have low density and poor mechanical properties. With the iteration of lasers [19], SLM [20,21], which uses high-energy lasers to melt all metal powders and directly solidify them into shape, has become mainstream. Parts formed by SLM not only have good mechanical properties but also extremely high forming accuracy. EB-PBF utilizes an electron beam to strike metal powder, instantly converting its kinetic energy into thermal energy and then melting and shaping it [22,23]. Although its forming accuracy is slightly lower than that of SLM, oxidation does not occur in a vacuum environment. L-DED [24,25,26,27] and EB-DED [28,29] rely on driving devices to supply powder/wire. Once the material is delivered to the designated position, the laser/electron beam focuses to generate a molten pool. The position of the molten pool has a large degree of freedom and has unique advantages in the field of part repair. Compared with PBF technology, DED has a faster forming speed, especially WA-DED formed by arc melting, which can reach several kilograms per hour, and is particularly suitable for the manufacturing and processing of large parts [30,31]. However, its surface quality is negatively correlated with forming accuracy and forming speed. Therefore, when choosing MAM technology, a comprehensive assessment is needed to select the best option.

### 1.2. In-Process Monitoring Technologies for Additive Manufacturing

Owing to its exceptional design freedom, capability for fabricating complex geometries directly, short production cycles, and high material utilization efficiency, AM is poised to transform the manufacturing landscape, emerging as a viable alternative to traditional subtractive methods [32]. However, the widespread adoption of AM is impeded by concerns over process reliability. Process anomalies can lead to safety hazards, while internal defects (e.g., porosity and cracks [33]) within as-fabricated parts critically compromise their mechanical performance and service life. These challenges underscore the critical need for robust in-process monitoring. Currently, most production-grade AM systems lack integrated monitoring capabilities. The integration of sensor-based monitoring systems, enabling real-time tracking of machine condition and part quality with closed-loop feedback, is therefore paramount for enhancing process stability and ensuring product quality [34].

At present, there are various technologies mainly practiced for online monitoring in the field of AM, as shown in Figure 2, including visual monitoring, ultrasonic monitoring, radiographic monitoring, thermal imaging monitoring, and acoustic emission monitoring, etc.

Visual monitoring employs high-speed cameras to capture the printing process, enabling defect analysis based on image information [35,36]. For instance, Ito et al. [37] utilized high-precision cameras to acquire inter-layer images after metal powder melting in LB-PBF, calculating surface porosity. Similarly, Xia et al. [38] monitored the melt pool morphology evolution during WA-DED using optical cameras, establishing correlations between melt pool characteristics and various defect types. While visual monitoring offers the most intuitive feedback, it is generally limited to detecting surface-level defects, and its resolution constrains the analysis of minute features [39].

Ultrasonic monitoring involves transmitting ultrasonic waves into the material and correlating the attenuation of reflected waves with internal defects [40]. Yang et al. [41] demonstrated the feasibility of using ultrasound to probe the metal melt pool, showing its sensitivity to defect formation during dynamic processes. Raffestin et al. [42] successfully identified and localized defects in LB-PBF by analyzing low-amplitude ultrasonic echoes.

Radiographic monitoring utilizes penetrating radiation (e.g., X-rays) to scan the material, where variations in grayscale in the transmitted image reveal the location and size of internal defects [33]. Mireles et al. [43] applied computed tomography (CT) to identify defects in EB-PBF fabricated parts, while Everton et al. [44] used radiographic scanning to distinguish parts with different porosity levels produced by LB-PBF. Both ultrasonic and radiographic techniques enable the inspection of internal defects; however, their effectiveness is influenced by material penetration depth, and they exhibit lower sensitivity to defects oriented parallel to the direction of wave or radiation propagation [45,46].

Thermographic monitoring uses thermocouples or infrared sensors alongside thermography to reconstruct the melt pool’s shape and size. It correlates temperature variations in 3D thermal maps with the defect formation process [47]. Ding et al. [48] generated thermal images from infrared data to analyze defect generation in L-DED, and McNeil et al. [49] investigated the propagation of cold cracks during LB-PBF by monitoring temperature gradients with infrared sensors. This approach provides an intuitive thermal perspective of the AM process. Nevertheless, the substantial computational load required to construct detailed thermal images poses a challenge for direct application in real-time monitoring [48].

The AE online monitoring technology analyzes the sound wave signals generated during the material processing to provide an efficient monitoring method for the AM process. This technology does not require integrating sensors into the processing head, and it is flexible in deployment, enabling low-cost and real-time online process monitoring [50]. In practical applications, Raeker et al. [51] utilized piezoelectric transducer (PZT) sensors and AE technology to achieve in situ crack detection in a single-channel laser melting experiment. By analyzing the acoustic emission signals, they revealed the influence patterns of different laser powers and scanning speeds on crack size, as well as the crack initiation characteristics of the material. Ansari et al. [52] proposed a monitoring method based on the exponential decay characteristics of AE signals. Through the analysis of the second derivative, noise was effectively filtered out, enabling the identification, quantification of surface and internal cracks during DED, as well as the reliable correlation of their initiation time and location. Furthermore, Xu et al. [53] combined AE technology with machine learning algorithms to systematically analyze the influence of key process parameters such as overlap rate, dwell time, and layer number on the relationship between AE features and Rockwell hardness. By integrating strain sensors, they achieved real-time and non-destructive prediction of the hardness of L-DED components, demonstrating a promising application in engineering. However, Harke et al. [54] also conducted research on the relationship between the surface acoustic wave (SAW) signals generated by laser ultrasonic waves and the surface and subsurface defects during the L-PBF forming process.

AE monitoring detects elastic waves generated within the material during fabrication, allowing for real-time assessment of the process dynamics. Hossain et al. [55] investigated the progress of acoustic technology in modulation processes and part quality monitoring, and also explored the potential applications of acoustic technology in quality inspection and monitoring of modulation techniques. Compared with the existing reviews, the novelty of this review does not merely lie in the simple listing of acoustic emission technology. Instead, it lies in constructing for the first time a deep analysis framework that runs through the entire technical chain of “hardware configuration—parameter correlation—intelligent monitoring—closed-loop control”. It deeply analyzes the unique challenges and solutions faced by acoustic emission technology from signal acquisition and processing to the final realization of intelligent decision-making, filling the gap of the lack of specialized and systematic reviews in this field.

## 2. Acoustic Emission System and Its Feature Extraction Methods

AE refers to the phenomenon where a material rapidly releases localized strain energy in the form of transient elastic waves upon deformation or fracture under external or internal stresses [56,57]. During the additive manufacturing process, anomalous events such as process instability or the formation of internal defects (e.g., pore generation and crack propagation) act as sources of AE, generating elastic waves with distinct characteristics [58,59,60]. By applying specific parameter extraction methods to isolate these signal features and correlating them with specific process dynamics, AE technology serves as an effective tool for in-process monitoring of MAM.

The basic composition of a sound emission monitoring system is shown in Figure 3. This system mainly consists of an AE source, an AE sensor, an amplifier, a signal processing system, and a recording and display system. During operation, the elastic waves generated due to defects or abnormalities during additive manufacturing are captured by the AE sensor, then amplified by the amplifier, and further analyzed and processed by the signal processing system. Finally, the results are transmitted to the recording and display system to achieve monitoring of the metal additive manufacturing process.

### 2.1. Acoustic Emission System Configuration

The composition of the acoustic emission system is shown in Figure 4, mainly including an AE sensor, a preamplifier, a filter, an AE signal acquisition instrument, and a computer with dedicated software. The AE sensor is coupled to the material under test. The remaining components are connected in sequence via coaxial cables to form a complete signal acquisition and processing chain.

The AE sensor, which is directly or indirectly coupled to the workpiece, serves as the core component of an AE system for data acquisition. It converts mechanical vibrations (elastic waves) from the workpiece into electrical signals. The classification of AE sensors is shown in Figure 5. Based on the wave propagation medium, AE sensors are broadly categorized into air-borne acoustic emission (ABAE) and structure-borne acoustic emission (SBAE) sensors [61,62].

As the name suggests, ABAE sensors detect AE signals transmitted through air vibrations. The most common ABAE sensor is the microphone, which offers ease of deployment but generally suffers from limited accuracy. In contrast, SBAE sensors capture waves propagating through the structure itself. Primary SBAE sensors include piezoelectric transducer (PZT) AE sensors and optical fiber (OF) AE sensors. The latter can be further classified into fiber Bragg grating (FBG) AE sensors, fiber optic ring (FOR) AE sensors, and Fabry–Perot interferometric (FPI) AE sensors based on their distinct sensing principles [63]. PZT AE sensors [64,65] utilize piezoelectric ceramics to transduce elastic waves into electrical signals. They are characterized by high sensitivity, excellent stability, and low cost. However, their performance can be compromised in harsh environments involving strong electromagnetic interference, significant corrosion, or elevated temperatures [66]. FBG sensors operate by detecting shifts in the reflected Bragg wavelength induced by strain variations from elastic waves [67]. The primary detection mechanism of the FBG sensors is not based on changes in the refractive index of the surrounding air. Instead, the high-frequency sonic/ultrasonic waves generated by the build process physically impinge on the optical fiber, inducing dynamic strain within the fiber itself. This strain directly alters both the period of the grating (Λ) and the effective refractive index (n) of the fiber core through the photo-elastic effect, resulting in a measurable shift in the Bragg wavelength $\left(\lambda_{B}\right)$. The air serves as the propagation medium for the acoustic energy, but the FBG fundamentally responds to the resulting mechanical deformation.

They offer high sensitivity and are particularly notable for their small size and potential for embedment within structures, garnering significant research interest [68]. A limitation is their direction-dependent sensitivity, which may restrict application in certain scenarios. FOR sensors [63] incorporate a fiber-optic ring mirror into the FBG configuration, enabling the simultaneous detection of AE signals from multiple directions. FPI sensors function by measuring changes in the cavity length of a FPI, which alters the interference pattern of light. This design provides a wide measurement range and high stability [69,70], albeit often at the cost of complex and challenging manufacturing processes.

The preamplifier boosts the weak AE signals from the manufacturing process to a sufficient amplitude via an amplification circuit. This ensures reliable long-distance transmission and proper reception by subsequent instruments. The filter then removes environmental interference and current noise introduced during cable transmission by applying predefined frequency thresholds. Serving as the core processing unit (“key”) of the AE system, the signal acquisition instrument performs two critical functions: analog-to-digital conversion of the signals and extraction of characteristic AE parameters. The methodologies for feature parameter extraction will be the focus of the next section. Finally, the computer stores these parameters and, with the aid of specialized AE analysis software, facilitates model development by establishing correlations between the AE features and the AM process dynamics, thereby enabling in-process monitoring capabilities.

### 2.2. Acoustic Emission Feature Extraction

#### 2.2.1. Time-Domain Analysis

The raw signal captured directly by an acoustic emission system is a time-domain waveform. Through analysis of this waveform, characteristic parameters can be extracted to quantify the signal’s features. These include fundamental descriptors such as amplitude, energy, ring-down count, duration, and root mean square (RMS) value. Further parameters encompass average signal level (ASL), threshold voltage, rise time, peak count, signal strength, absolute energy, and time of arrival.

#### 2.2.2. Frequency-Domain Analysis

Frequency-domain analysis primarily involves spectral analysis, energy spectrum analysis, power spectrum analysis, and cepstrum analysis. Transforming temporal features into the frequency domain enables a more effective analysis of the information embedded within transient waveforms. Typically, spectral analysis refers to the examination of signals via Fourier transform (FT), which includes both magnitude and phase spectra; however, the magnitude spectrum is most commonly utilized in practice.

The Fourier transform of a continuous signal *x*(*t*) is defined as follows [71]:(1)$$X\left(w\right)=\int_{-\infty }^{+\infty }x(t)e^{-j\omega t}dt$$

The Discrete Fourier Transform (DFT) of a finite-length discrete signal is defined as follows [72]:(2)$$X\left(k\right)=\sum_{n=0}^{N-1}x\left(n\right)W_{N}^{k_{N}},\;k=0,1,\ldots ,N-1;W_{N}=e^{-j\frac{2\pi }{N}}$$

The energy spectrum, also referred to as the energy spectral density (ESD), describes the distribution of signal energy across the frequency domain. The power spectrum, fully termed the power spectral density (PSD) function, represents signal power per unit frequency band. Let s(t) be an energy signal. The formulae for its ESD and PSD are given as follows:

Energy Spectral Density:(3)$$E=\int_{-\infty }^{+\infty }s{\left(t\right)}^{2}dt=\int_{-\infty }^{+\infty }{\left|S(f)\right|}^{2}df$$

Power Spectral Density:(4)$$P\left(f\right)=\underset{T\to \infty }{\lim}\frac{1}{T}\left|{S(f)}^{2}\right|$$

Cepstrum, also referred to as the second-order spectrum or logarithmic power spectrum, is an analytical technique that facilitates the extraction and identification of periodic components in a signal that are difficult to discern visually in the original spectrum. It simplifies families of sideband spectral lines in the conventional spectrum into individual spectral lines and is less influenced by sensor location and transmission path effects. The cepstrum is mathematically defined as the inverse FT of the logarithm of the power spectrum, which can be summarized by the following procedure: original signal → compute power spectrum → take logarithm → apply inverse FT.

Commonly used frequency-domain feature parameters include centroid frequency, mean frequency, root mean square frequency, frequency variance, energy spectral density, and power spectral density.

#### 2.2.3. Wavelet Analysis

Wavelet transform provides localized analysis in both time (or space) and frequency domains [73,74]. It progressively refines a signal through scaling and shifting operations, enabling multi-resolution analysis. This approach achieves high temporal resolution at high frequencies and high frequency resolution at low frequencies, thereby adaptively meeting the requirements for time-frequency analysis of non-stationary signals. Consequently, it allows for detailed examination of signal features at arbitrary scales. The continuous wavelet transform is defined as follows [75]:

For any function $\psi \left(t\right)\in L^{2}(R)$, ψ(t) is a square-integrable function, if its Fourier transform satisfies the admissibility condition:(5)$$\int_{-\infty }^{+\infty }\frac{{\left|\varphi (\omega )\right|}^{2}}{\omega }d\omega <\infty$$
then ψ(t) is termed a basic wavelet or mother wavelet. A valid mother wavelet must fulfill the following requirements:Normalization (Unit Energy):(6)$$\int_{-\infty }^{+\infty }{\left|\varphi (t)\right|}^{2}dt=1$$

2.Boundedness:


(7)
$$\int_{-\infty }^{+\infty }\left|\varphi (t)\right|dt<\infty$$


3.Zero Mean (Admissibility Condition):


(8)
$$\int_{-\infty }^{+\infty }\varphi (t)dt=0$$


#### 2.2.4. The Short-Time Fourier Transform

Unlike wavelet group decomposition, the short-time Fourier transform (STFT) analysis avoids the influence of low-amplitude signals on the frequency domain defect signals. By dividing the signal into small time windows and performing FT on each window, STFT analysis can extract the frequency components of each moment of the corresponding defect signal, thereby obtaining the frequency characteristics of the defect signal over time [76]. Assuming the time-varying signal is x(t), and the window function is g(t), the STFT is defined as:(9)$$STFT\left(t,f\right)=\int_{-\infty }^{+\infty }x(n)g^{*}(t-\tau )e^{-j\omega \tau }dt$$

*t* represents the corresponding time, f is the signal frequency, τ is the center position of the window, $g^{*}(t-\tau )\;$is the complex conjugate of the time window function g(t−τ). The discrete FT is used to process signals, and its form is:(10)$$STFT\left(n,f\right)=\sum_{n=-\infty }^{\infty }x(n)g^{*}(n-m)e^{-j2\pi fm}dt$$

*x*(*n*) is a discrete AE signal, g(m) is a window function.

## 3. Applications of Acoustic Emission for In-Process Monitoring of Metal Additive Manufacturing

### 3.1. Hardware Configuration for AE Monitoring

In the in-process monitoring of metal additive manufacturing using AE, different types of AE sensors exhibit distinct performance characteristics, with installation position playing a decisive role in signal strength and various forms of noise significantly affecting measurement accuracy. Therefore, the design of AE monitoring hardware focuses on three critical aspects: sensor selection, optimal placement, and noise suppression.

The microphone uses air as the transmission medium and does not need to come into direct contact with the work area. Common installation positions are shown in Figure 6. It can be fixed around the work area through a bracket or welding [77,78,79,80]. However, in L-PBF systems, their installation must maintain the airtight integrity of the build chamber. The primary limitation of microphones lies in their relatively low intrinsic sensitivity, compounded by strong signal attenuation in air, which further reduces the captured AE signal amplitude. In contrast, FBG sensors offer high sensitivity by detecting vibrations through changes in refractive index, ensuring accurate signal acquisition. For instance, Shevchik et al. [81,82] monitored the build region in SLM by affixing FBG sensors to the chamber wall using plastic supports, as shown in Figure 7.

Nevertheless, most researchers continue to favor PZT sensors due to their operational stability and cost-effectiveness. Mounting PZT sensors directly on or beneath the substrate allows for the capture of comprehensive and clear process signals.

The installation position of the PZT sensor in the additive manufacturing process is shown in Figure 8. The principle of the AE PZT sensor is based on the positive piezoelectric effect. It converts the transient elastic stress waves (AE signals) generated due to damage (such as crack propagation, fiber fracture) within the material into electrical signals. The detection object selects a sensor with an appropriate resonant frequency and sensitivity and ensures excellent acoustic coupling between the sensor and the surface of the component by using a coupling agent and a constant clamping force. In application, setting reasonable threshold values and parameter analysis are used to distinguish real AE events from noise. At the same time, it must be calibrated with standard AE sources (such as broken lead) to ensure the accuracy of data acquisition and the repeatability of the results. However, in PBF applications, routing the sensor cables without interfering with the powder recoating mechanism remains a challenge [37,83,84]. In DED processes characterized by high heat input [85,86,87], particularly in WA-DED [88,89], elevated temperatures pose a risk of sensor damage. While placing sensors underneath the WA-DED substrate can protect them from thermal exposure, this configuration significantly attenuates signal strength due to damping through the platform and base material [90].

In response to the aforementioned challenges, the following solutions are proposed in this review: For PBF systems, a recess can be machined beneath the build platform to accommodate a PZT sensor, as shown in Figure 9a, accompanied by a dedicated channel for cable routing. This configuration enables flexible cable management without interfering with the powder recoating process. For DED systems, particularly in WA-DED, a waveguide fabricated from a low-acoustic-attenuation material can be welded to the substrate, as shown in Figure 9b. Alternatively, an integrated water-cooling system can be applied to the substrate, as shown in Figure 9c. Both approaches effectively reduce the operational temperature of the sensor, ensuring its reliability and longevity.

Existing denoising techniques, including wavelet denoising in MATLAB (v. 2021) [83], homomorphic deconvolution [85,86], and digital filters based on local least squares methods [91], have proven effective in enhancing signal quality and preserving data accuracy in monitoring processes such as LB-PBF and L-DED [37,77,81,82,84,91,92]. However, WA-DED presents a more challenging scenario due to its intense and diverse noise sources, which can often overwhelm signals originating from the deposition zone. Conventional filtering methods are only capable of removing low-amplitude interference in such environments. A more targeted and practical approach for WA-DED involves identifying the characteristic frequencies of dominant noise sources and systematically eliminating these frequency components from the spectrum. This strategy has been demonstrated as the most effective solution currently available for WA-DED applications [78,79].

### 3.2. Correlation of AE Parameters with Process Dynamics and Internal Defects

Effective signal denoising is a necessary condition for accurately extracting feature parameters from the data. Distinct feature parameters extracted from acoustic emission signals via various signal processing techniques can be employed to characterize both the additive manufacturing process—including the effects of process parameters, powder melting/droplet transfer, spattering, and anomalous states—and internal part defects, primarily porosity and cracks.

#### 3.2.1. Process Characterization

##### LB-PBF AE Process Characteristics

Uniform powder melting and stable droplet transfer are essential for producing parts with superior mechanical properties. In LB-PBF, the volumetric expansion during powder melting and subsequent contraction upon solidification compresses or displaces the surrounding atmosphere, generating detectable AE signals. Hauser et al. [78] demonstrated that AE signal peaks can effectively identify abnormal melting conditions in laser metal deposition (LMD), such as those resulting from excessive heat accumulation and corresponding changes in melt pool geometry.

As an inevitable by-product of LB-PBF, spatter originates from the ejection of molten particles around the melt pool. Larger spatter particles often fail to fully remelt under laser irradiation, which induces slag inclusion and pore defects in the fabricated components [93]. Additionally, spatter particles that settle back into the powder bed disrupt powder layer uniformity, thereby further deteriorating the overall quality of the final parts [94].

Spatter is primarily caused by the rapid release of energy during the melting of metal powder under high-energy beams. Partial vaporization of molten metal and the resultant recoil pressure generate a vapor plume, which expels molten droplets and partially melted particles, resulting in spatter [95,96,97,98,99]. AE-based monitoring of spatter essentially detects air vibrations around the melt pool. As spatter occurs, the density of the vapor plume increases with rising melt pool temperature, causing pressure fluctuations and exciting AE signals through air vibration. During the melting process, the acoustic emission equipment monitors the acoustic signals of the melted and non-melted materials in the time domain and frequency domain. Through time-domain and frequency-domain analysis, noise can be removed and the frequency range of the melting acoustic signals can be determined. At the same time, different agglomeration and rupture phenomena generated under different melting conditions are analyzed [80].

Several studies have demonstrated the capability of AE in quantifying spatter behavior. For instance, Luo et al. [100] found that in the resistance spot welding process of aluminum alloys, acoustic emission signals provide a direct characterization of the physical stage of nucleation. Pal et al. [101] identified AE frequency as a distinguishing feature for spatter intensity during droplet transfer in gas metal AM. In SLM of 316 L stainless steel, Luo et al. [77] induced different spatter states by varying laser power and applied short-time Fourier transform to AE signals. They identified a peak at 25 kHz that effectively distinguished different levels of spatter, as shown in Figure 10.

##### L-DED AE Process Characteristics

Similarly, improper powder/wire feed rate—either too high, causing incomplete melting, or too low, leading to overmelting—also induces distinct AE responses [102,103,104,105]. Whiting et al. [92] detected powder flow velocity in L-DED by analyzing AE signals generated from powder impacts on the feed channel, showing that the root mean square (RMS) value serves as a reliable indicator, as shown in Figure 11.

Monitoring fluctuations in the working state provides a more direct means of assessing the overall additive manufacturing process, as any process anomalies ultimately manifest as variations in these states. Hossain et al. [85] demonstrated that after wavelet processing and Fourier transformation of the acquired signals, ratios such as peak amplitude (PA) to peak frequency (PF) and centroid amplitude (CA) to PF effectively distinguish different working states in L-DED, including system off, powder feeding only, optimal process conditions, low laser power, and low powder feed rate. Taheri et al. [86] utilized high- and low-frequency characteristics in the spectrum to identify different operational states in L-DED, while Hauser et al. [78] found that abrupt changes in the average intensity of the Mel spectrum can reliably indicate sudden anomalies during otherwise stable L-DED operation, as shown in Figure 12.

Although AE-based monitoring of state anomalies offers an intuitive indication of overall process stability, it generally cannot identify the specific root cause of the anomaly. A comprehensive diagnosis requires integration with other monitoring perspectives to pinpoint issues and implement corrective measures.

##### WA-DED AE Process Characteristics

Process parameters form the foundation of AM, whose appropriateness directly determines the rationality of the process outcome [106,107,108,109,110]. The AM process is invariably accompanied by the flow of shielding gas, where excessive flow rate may disrupt the melt pool, while insufficient flow can lead to severe oxidation [111]. For instance, Hauser et al. [78] suggested that the AE source in WA-DED originates from arc-induced plasma expansion and used Mel-frequency AE signal intensity to characterize arc behavior under different shielding gas flow rates. Zhu [88] and Luo et al. [89] observed that each droplet transfer event during WA-DED corresponds to a distinct peak in the time-domain AE signal. By analyzing the periodicity of these peaks, different WA-DED modes can be effectively identified, as shown in Figure 13. Furthermore, Hauser et al. [90] distinguished the melting behavior of different aluminum alloy wires by analyzing AE frequency shifts when switching between CMT and CMT+P modes. Hauser et al. [90] established a correlation between AE signal intensity and wire feed speed, as shown in Figure 14.

#### 3.2.2. Characterization of Internal Defects in Fabricated Parts

Internal defects such as porosity and cracks significantly compromise the service performance and lifespan of additively manufactured components. Monitoring the formation of these defects using AE technology is therefore of considerable importance for quality assurance.

Porosity typically originates from the entrapment of metal vapor within the melt pool during rapid melting and evaporation. Influenced by heat transfer, fluid flow, the Marangoni effect, vapor recoil pressure, and keyhole instability, the vapor cannot escape promptly and becomes trapped at the bottom of the melt pool, resulting in pore formation upon solidification [112,113]. AE monitoring captures this phenomenon by detecting the elastic waves emitted as the trapped vapor oscillates within the molten material.

In contrast to porosity, crack formation in AM involves more complex mechanisms, and AE technology also exhibits high effectiveness in capturing crack-related signals. Three principal mechanisms are currently recognized to explain crack formation in additive manufacturing: (1) Solid-state cracking occurs when residual thermal tensile stresses exceed the material’s ultimate tensile strength during the process [114,115]; (2) Solidification cracking arises from the rupture of liquid films within the mushy zone as metal fluidity decreases toward the end of solidification [116,117,118]; and (3) Intergranular cracking results from carbide liquefaction in certain alloys, forming low-melting-point eutectic films along grain boundaries [119,120]. Researchers have established that crack propagation emits AE signals with distinctive frequency characteristics regardless of the cracking mechanism [121], making frequency-domain analysis a viable approach for monitoring crack initiation and growth during AM. Mohammadi et al. [83] demonstrated that the spectral distribution of AE signals can differentiate between cracks and porosity by fabricating specimens with controlled defects through variation of SLM laser power, as shown in Figure 15. The signal characteristics corresponding to each defect are different. Figure 15 (a-c), respectively, represent the minimum defect (class 1), crack defect (class 2), and porosity (class 3). Similarly, Ramalho et al. [79] introduced artificial defects during WA-DED and observed corresponding shifts in AE frequency components when the deposition passed through these regions.

##### LB-PBF AE Defect Monitoring

AE technology has emerged as a non-destructive and real-time monitoring method for characterizing the formation and evolution of defects during additive manufacturing processes. Several studies have demonstrated the correlation between AE features and porosity. Wang et al. [84] fabricated 316 L stainless steel samples with varying porosity levels by adjusting laser power in SLM. After processing the AE signals with Adaptive Ensemble Entropy—Whale Optimization Algorithm—Variational Mode Decomposition (AEE-WOA-VMD), as shown in Figure 16a, they found that power spectral density and energy entropy effectively distinguished different porosity levels, as shown in Figure 16b.

In a study, Shevchik et al. [81,82] decomposed AE signals using a Daubechies wavelet with ten vanishing moments and proposed the relative energy in specific narrow frequency bands as a discriminative feature for porosity. While existing AE methods show promise in classifying parts with different porosity levels, current research remains largely focused on macroscopic porosity assessment. A precise AE feature that directly corresponds to the dynamic process of pore formation has not yet been firmly established. Identifying such a feature represents a critical direction for future research.

Beyond frequency analysis, time-domain AE parameters have also proven effective in crack detection. By correlating AE signals with micro-computed tomography, Seleznev et al. [91] showed that time-domain features such as RMS, standard deviation, and peak amplitude can characterize micro-crack formation during SLM fabrication and cooling, with AE energy levels even enabling crack size differentiation, as shown in Figure 17. Two crack signals appeared at the 5th and 8th minutes of the manufacturing process. They seemed to correspond to the cracks near the substrate at the cutting site. The first crack (at a height of 4 mm) appeared to be associated with the AE signal recorded at the 18th minute. This indicates that crack defects can be monitored through the time-domain characteristics of acoustic emission signals. Ito et al. [37] further utilized AE signal arrival times to locate the origin of pore or crack formation. Compared to porosity monitoring, AE technology shows greater potential for crack characterization in AM. Characteristic AE parameters not only identify the onset of crack propagation but also show promise in estimating crack dimensions, owing to the relatively prolonged dynamic nature of crack extension.

##### L-DED AE Defect Monitoring

AE technology has emerged as an efficient method for defect monitoring in the L-DED process, owing to its capability capturing the dynamic response characteristics of materials in real time. In the L-DED process, acoustic wave vibrations are excited by the interaction between high-energy lasers and materials, thereby generating AE signals. Li et al. [122] used AE sensors to collect the AE signals and signal energy during the deposition process, as shown in Figure 18. It can be observed that the AE signals during the normal deposition process are relatively smooth, while the signal energy amplitude at the crack location is significantly greater than that in the normal state. Wang et al. [123] combined AE monitoring technology with finite element analysis to investigate the initiation position and expansion range of cracks in the L-DED process, confirming that the number of cracks is positively correlated with the coating area, thickness, and cooling rate. Strantza et al. [124] used AE technology to monitor the crack propagation behavior during the deposition process and established a correlation model between AE characteristic parameters and the initiation position of cracks. Gaja et al. [125] collected AE signals during the deposition of mixed metal powder and used K-Means clustering, logistic regression (LR), and artificial neural network (ANN) models to accurately identify AE signals of pores and cracks; the study found that AE signals corresponding to pores have a short attenuation time and small amplitude, while the signals induced by cracks exhibit short duration and high amplitude characteristics.

##### WA-DED AE Defect Monitoring

In response to the actual demands of shortening the production cycle of arc-based AM and improving the forming quality, many researchers have introduced the AE non-destructive testing technology into this field to achieve real-time and automated monitoring of defects during the deposition process. Zhang et al. [126] used the detection data collected by the ultrasonic sensor to not only reflect the forming state of the deposition layer but also accurately identify the surface discontinuity defects and their location information in the weld bead metal. As shown in Figure 19a, there is a concave defect on the weld bead surface; the corresponding average signal level (ASL) scan image (Figure 19b) and the ultrasonic absolute energy distribution graph (Figure 19c) both show obvious signal mutation characteristics, from which the location of this surface discontinuity defect can be precisely located. Research shows that such deposition non-uniformity phenomena or the generation of deposition defects will cause a sharp increase in the absolute energy of the ultrasonic wave.

Table 1 summarizes the application of AE technology for the online monitoring of MAM, with a focus on the L-PBF, L-DED and WA-DED processes. The primary defects detected are pores and cracks. For AE signal processing, the table outlines a range of techniques, spanning from traditional methods to intelligent algorithms. These include classical signal processing approaches, such as digital noise filtering and Welch power spectrum estimation, alongside machine learning algorithms like the K-Means Clustering algorithm (K-Means) and Gaussian Mixture Models (GMM). Furthermore, the summary incorporates advanced deep learning models, including Spectral Convolutional Neural Networks (SCNN) and Variational Autoencoders (VAE), as well as hybrid strategies like AEE-WOA-VMD.

### 3.3. Intelligent In-Process Monitoring

#### 3.3.1. Development and Application of Classification Models

Although AE feature parameters can characterize both the AM process and the quality of the resulting parts, the vast volume of acquired AE data makes manual identification and correlation both time-consuming and prone to inaccuracies. It is therefore essential to develop classification models for the automated categorization of AE data. Commonly employed methods for constructing such models include polynomial regression equations [127,128], fruit fly optimization algorithms [129,130], K-means clustering [131,132], neural networks [133,134], random forests [135,136], and support vector machines [137].

##### Polynomial Regression

Polynomial regression is a supervised learning algorithm in machine learning. Its core principle is to fit a polynomial equation based on historical data and utilize this equation to predict outcomes for new data points [138]. Polynomial regression models respond quickly and deliver high monitoring accuracy with moderately sized datasets. In a study by Whiting et al. [92], such a model, using RMS values as the feature parameter, effectively monitored powder flow rates during laser-based directed energy deposition. After filtering interfering signals, it detected abrupt flow rate changes with 95.9% accuracy for 316 L powder and 98.5% for commercially pure titanium.

##### Fruit Fly Optimization Algorithm

The Fruit Fly Optimization Algorithm (FOA) is an emerging swarm intelligence optimization method inspired by the foraging behavior of fruit flies. It performs a population-based iterative search through the solution space to locate optimal solutions [139]. It performs a population-based iterative search through the solution space to locate optimal solutions. The general process of the FOA is as follows: First, randomly initialize the positions of the fruit fly population and assign a random flight direction and distance to each fruit fly. Then calculate the odor concentration determination value, substitute it into the objective function to calculate the odor intensity, and screen the optimal individual. Finally, the fruit fly population moves towards the optimal individual position. Repeat the above steps until the number of iterations is exceeded or the optimal solution meets expectations.

##### K-Means Clustering Algorithm

K-Means Clustering Algorithm (K-Means) is an unsupervised iterative clustering analysis method in machine learning that partitions a given dataset into K clusters by minimizing the within-cluster sum of squared distances [140]. It is primarily used to automatically group similar samples into the same category. Its principle lies in starting with randomly initialized cluster centers, assigning each data point to the nearest cluster, and then calculating the mean of the points within each cluster to serve as the new cluster centers. This iterative process continues until the cluster centers converge. Building upon the fundamental principles of K-Means clustering, researchers have successfully implemented this algorithm for defect identification and process monitoring in AM.

##### Neural Networks

Neural Networks (NN), also known as Artificial Neural Networks (ANN), are widely used for classification and recognition tasks. This method usually requires collecting a large amount of data and labeling it to create a dataset. Then, a neural network model is built. After a long period of training, the network model will have a certain generalization ability and can achieve the expected classification or detection effect on unknown data. The strength of neural networks lies in their ability to autonomously learn and adapt to data during training, making them highly popular for recognition applications. Luo et al. [77] used a convolutional neural network (CNN) with acoustic inputs and spatter images to classify different levels in L-DED, achieving over 85.08% accuracy. Wang et al. [84] applied a neural network with AEE-WOA-VMD features to predict SLM part quality, reaching 83.3% accuracy.

##### Support Vector Machines

Support Vector Machines (SVM) represent a supervised learning algorithm in machine learning, widely applied to statistical classification and regression analysis. The SVM algorithm first maps the low-dimensional input space to a high-dimensional feature space through a kernel function, constructs an objective function with the property of a convex function to transform the problem to be solved into a quadratic programming problem, and uses optimization methods such as stochastic gradient descent to find the optimal classification hyperplane. A key advantage of SVM lies in its ability to construct a robust high-dimensional model even with limited sample data. Wang et al. [84] used AE features with SVM for quality assessment in SLM, achieving 91.7% recognition accuracy—significantly higher than a neural network on the same dataset (83.3%). In another application, Wu et al. [141] applied SVM to monitor the material extrusion process in fused deposition modeling (FDM) printing, using AE energy features to identify five extrusion states with 95% accuracy.

##### Random Forest

Random Forest (RF) is an ensemble learning method. It consists of a forest of decision trees, each of which is trained on a “re-sampled” subset of the dataset. During the decision-making process, each tree classifies the sample, and the final classification result is determined by the majority voting method. The key advantage of RF is its ensemble method, which often yields higher accuracy than individual algorithms. While RF has not yet been applied to AE in-process monitoring of additive manufacturing, it shows strong potential based on related applications. For instance, Zhang et al. [142] used RF with multiple AE features to predict fatigue crack propagation under extreme conditions, achieving 97.6% accuracy—demonstrating its promise for AE-based monitoring. This success demonstrates the significant potential of RF for AE-based in-process monitoring applications.

The performance variations are attributed to a combination of factors, primarily including the inherent complexity and spectral overlap of AE signatures from certain physical phenomena, which challenge the feature extraction capability of simpler models. Furthermore, the performance of some models was constrained by the limited quantity of high-quality training data available for specific, rare defect types, leading to insufficient feature learning. The suboptimal performance of conventional machine learning models can also be linked to their dependence on hand-crafted features and their limited capacity to capture the complex, non-linear patterns in raw waveform data compared to deep learning architectures.

Table 2 demonstrates the performance of different modeling algorithms for online monitoring of AE in MAM, confirming the technology’s effectiveness in quality assurance. Each algorithm excels in specific tasks: Linear Regression achieves 98.5% accuracy in predicting powder flow rate for L-DED, showing high efficiency in linear problems; K-Means maintains 83–90% accuracy in defect identification and spatter classification, making it suitable for unsupervised scenarios; NN performs excellently in process state recognition with 96.0% accuracy, highlighting their capability to handle non-linear features. While applied in non-core scenarios, SVM and RF also achieve high accuracy rates of 91.7% and 97.6%, respectively, indicating strong generalization ability. The results show that by combining the appropriate algorithm strategy, AE monitoring can effectively diagnose key quality issues in the manufacturing process, providing a reliable basis for intelligent control.

#### 3.3.2. Closed-Loop Control with Acoustic Emission Monitoring

Closed-loop control (also referred to as feedback control) is a control strategy in which a portion or the entirety of a system’s output is fed back to its input through specific methods and devices [143,144,145]. This feedback signal is subsequently compared with the original input, and the resulting deviation is used to adjust the system’s operation, thereby preventing deviations from the predefined target. In the context of acoustic emission monitoring, this method continuously analyzes the acoustic emission signals and dynamically optimizes the process parameters of the manufacturing equipment for real-time control, in order to maintain the optimal forming state, as shown in Figure 20. Some researchers have already achieved substantial accomplishments in using machine learning based on these acoustic features for defect detection. Chen et al. [146] proposed a convolutional neural network based on Mel-Frequency Cepstral Coefficients (MFCC) features, Mel-Frequency Cepstral Coefficients convolutional neural networks (MFCC-CNN), for L-DED sound classification, aiming to detect cracks and keyhole pores. The overall accuracy rate reached 89%. This model outperformed traditional machine learning methods such as SVM and KNN in terms of keyhole pore recognition accuracy (93%) and AUC-ROC (98%). The study also indicated that noise removal of acoustic signals can effectively improve classification performance, which is of great significance for online monitoring and feedback control of acoustic emission.

Closed-loop control technology has now reached a relatively mature stage, with researchers having previously integrated visual and thermographic monitoring to implement such systems for additive manufacturing processes. Visual monitoring primarily tracks the melt pool length [147,148,149,150,151,152] and width [153,154], utilizing feedback algorithms to generate corrective instructions, thereby enhancing the quality of fabricated components. For instance, Xiong et al. [147] developed a control algorithm to automatically adjust the nozzle-to-substrate distance (NTSD) in GMAW, effectively regulating the melt pool height, as shown in Figure 21. Similarly, Zeinali et al. [148] employed a robust adaptive control algorithm to manage the melt pool height in LCF, both demonstrating effective control outcomes.

Thermographic monitoring, utilizing devices such as thermocouples and infrared pyrometers, focuses on the melt pool temperature and enables timely correction of energy input [149,151,155]. Song et al. [156] used a dual-color high-temperature meter to measure the molten pool temperature during laser cladding. The dynamic relationship between laser power and the molten pool was described using a state-space model. The temperature was experimentally identified using subspace methods. The closed-loop process can track the molten pool temperature to the reference temperature curve by adjusting the diode to control the laser power. The compensatory deficiency was verified using cladding for GPC, as shown in Figure 22.

In the domain of AE-based in-process monitoring, closed-loop control has been successfully implemented in traditional machining processes, where AE signals are utilized to effectively reduce tool wear [157,158,159]. However, to the best of our knowledge, the application of AE-based closed-loop control in additive manufacturing remains largely unexplored. The development of such a system for metal AM would involve several critical steps: extracting features from acquired AE signals, establishing classification models, and implementing real-time decision-making within a closed-loop monitoring framework. This system would enable real-time adjustments of process parameters for parts with minor defects, while triggering immediate shutdown procedures for severe defects or significant process anomalies. Advancing this intelligent AE monitoring capability represents a critical step forward in achieving fully intelligent metal additive manufacturing.

## 4. Perspectives and Future Directions

Currently, in situ monitoring of MAM based on AE technology is still in its early stages of development, and its research progress is closely related to the popularity of specific metal AM technologies. In the field of AE monitoring for additive manufacturing, the laser AM technique has received the greatest research attention due to its widespread application. Many researchers have proposed various interpretations of the correlation between AE signals and process dynamics; the WA-DED technology, with its significant advantages in large-part manufacturing, has also carried out targeted AE research; while the EB-DED technology has great potential, it is limited by high operating costs, and related AE monitoring research is relatively limited. In the future, the application of AE technology in the MAM process will evolve towards a more in-depth and systematic direction, with the ultimate goal of achieving true intelligent closed-loop manufacturing. Specific exploration and breakthroughs can be carried out from the following five core directions.

Construction of Multi-sensor Fusion and Intelligent Closed-loop Control System

Single sensing technology is unable to comprehensively capture the complex physical and chemical changes in the MAM process. AE technology needs to be deeply integrated with visual monitoring, thermal imaging monitoring, ultrasonic monitoring, etc., to construct a multi-sensor information network. Visual monitoring can accurately capture macroscopic dynamics such as the shape of the molten pool and the trajectory of splashing. Thermal imaging technology can obtain real-time temperature field distribution and gradient changes. Ultrasonic monitoring can assist in detecting internal deep defects, while AE technology is good at capturing transient signals of dynamic processes such as crack propagation and pore formation. Through multi-source data fusion algorithms (such as deep learning fusion models, Bayes theorem, etc.), cross-validation and complementary analysis of monitoring data from different dimensions can be carried out to achieve comprehensive and precise diagnosis of the manufacturing process status. On this basis, a multi-parameter adaptive collaborative control system needs to be established. Based on the fused feature information, key process parameters can be dynamically optimized to form a complete intelligent closed-loop of “perception-diagnosis-regulation”. This multi-sensor fusion-driven closed-loop control will significantly improve the stability of the manufacturing process and the consistency of product quality, and promote the transition of MAM from “passive monitoring” to “active regulation”.

2.Precise correlation between microstructure and AE signals

The current application of AE technology in MAM mostly remains at the level of statistical correlation, that is, by establishing statistical laws between acoustic emission characteristics and defect types, as well as process states through a large amount of experimental data. However, the interpretation of the microscopic physical mechanisms behind the signals is still insufficient. In the future, advanced in situ observation technologies such as synchrotron X-ray imaging and laser confocal microscopy need to be utilized to capture the microscopic physical processes during AM, such as pore nucleation and growth, crack initiation and propagation, molten pool flow and solidification, and powder melting. By matching and correlating these microscopic dynamics with the sequentially collected AE signals frame by frame, the generation mechanisms of AE signal characteristics (such as frequency, amplitude, energy, and duration) corresponding to different microscopic events can be revealed. This will enable AE technology to transform from a “phenomenon statistical tool” to a “mechanism analysis tool”, allowing for precise deduction of the evolution state of microscopic physical processes based on AE signals, achieving early and precise warning of defects and root cause location, and providing a more targeted theoretical basis for process optimization.

3.Establishment of the standardization system and development of the modular monitoring system

Currently, the application of AE monitoring technology in the field of MAM lacks a unified standard. Different research teams adopt significantly different sensor types, installation positions, signal acquisition parameters, noise suppression methods, etc., making it difficult to compare and reuse the experimental results, which severely restricts the industrialization promotion of the technology. In the future, it is necessary to accelerate the establishment of AE monitoring technology standards for different AM methods (such as L-PBF, WA-DED, EB-PBF, etc.). Clearly define the selection rules for sensors under different processes, the optimal installation positions and coupling methods, the setting standards for signal acquisition parameters, and the unified evaluation indicators and methods for noise suppression. At the same time, develop modular monitoring systems that are miniaturized, low-power, and have edge computing capabilities. This system should integrate core units such as sensor modules, signal preprocessing modules, data storage and computing modules, and communication modules, and can be flexibly assembled and adapted according to the model and process requirements of different AM equipment. Through edge computing technology, real-time preprocessing and feature extraction of AE signals can be achieved at the equipment end, reducing data transmission delay and meeting the real-time requirements of industrial sites; modular design makes it convenient for system maintenance, upgrading, and batch deployment, and reduces the cost of industrial application. In addition, develop standardized data analysis software platforms that integrate multiple signal processing algorithms and defect identification models, support visual analysis of data, historical data traceability, and generation of process optimization suggestions, and improve the practicality and usability of the monitoring system.

4.Expanding application scenarios and integrating with cross-domain technologies

Currently, the research on AE monitoring technology mainly focuses on laser and arc AM processes. In the future, the application scope needs to be expanded to emerging AM technologies such as EB-DED, BJ-AM, and MJ-AM. Additionally, cross-disciplinary technology integration needs to be strengthened, and cutting-edge technologies such as artificial intelligence, big data, and the internet of things should be introduced to promote the innovation and development of AE monitoring technology. For example, by combining the Transformer model in deep learning, the ability to extract features and improve defect recognition accuracy for long-time sequence AE signals can be enhanced; using big data technology to build a database of AE signals for MAM, including data of different materials, different processes, and different defect types, can provide data support for model training and process optimization; through the internet of things technology, the AE monitoring data of multiple AM devices can be networked and managed, enabling global process status analysis and quality traceability, and helping to facilitate the digital transformation of intelligent manufacturing production.

5.Improvement in extreme environment adaptability and monitoring capabilities of high-performance materials

As the application of MAM technology continues to expand in high-end equipment fields such as aerospace and nuclear industries, the manufacturing scenarios are facing increasingly complex challenges from extreme environments, including high temperatures, high pressures, strong electromagnetic interference, corrosive atmospheres, and other harsh conditions. At the same time, the application materials are gradually extending to high-temperature alloys, hard alloys, and composite materials, which pose higher requirements for quality monitoring in the manufacturing process. Therefore, it is necessary to break through core technologies specifically: First, focus on researching extreme environment-compatible AE sensors, such as high-temperature-resistant ceramic-based sensors, anti-electromagnetic interference optical fiber sensors, and corrosion-resistant encapsulated sensors, through material innovation and structural optimization, to enhance the stability and service life of the sensors in harsh conditions; Second, deeply explore the quantitative correlation laws between microstructural evolution such as grain growth, phase transformation, and second-phase precipitation during the AM of high-performance materials and AE signals. Given that the mechanical properties of high-performance materials are highly related to their microstructure, tracking the microstructural evolution in real time using AE signals can provide a new technical path for precisely optimizing process parameters and directionally regulating material properties; Third, develop signal processing algorithms specifically for extreme environments and high-performance materials, effectively filtering out environmental noise and interference caused by the material’s own characteristics, and further improving defect identification accuracy and the reliability of process monitoring.

In conclusion, the future development of AE technology in the field of in situ quality monitoring during MAM is a process of evolution from a single technology application to system-level intelligence, from statistical phenomena to the exploration of physical essence, from laboratory tools to industrial standard solutions, and continuous expansion of the application boundaries. Through continuous research and technological breakthroughs in the above directions, an intelligent AM system with self-sensing, self-diagnosis, self-decision-making, and self-optimization capabilities will eventually be constructed, providing core quality assurance for the production of high-reliability and high-performance metal components, and promoting the large-scale application of metal additive manufacturing technology in more high-end fields.

## Figures and Tables

**Figure 1 sensors-26-00438-f001:**
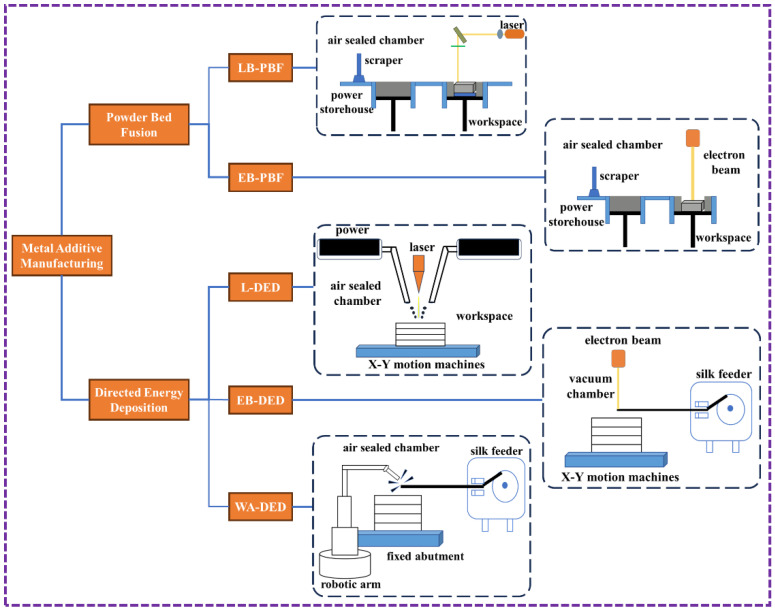
Classification of metal additive manufacturing technologies.

**Figure 2 sensors-26-00438-f002:**
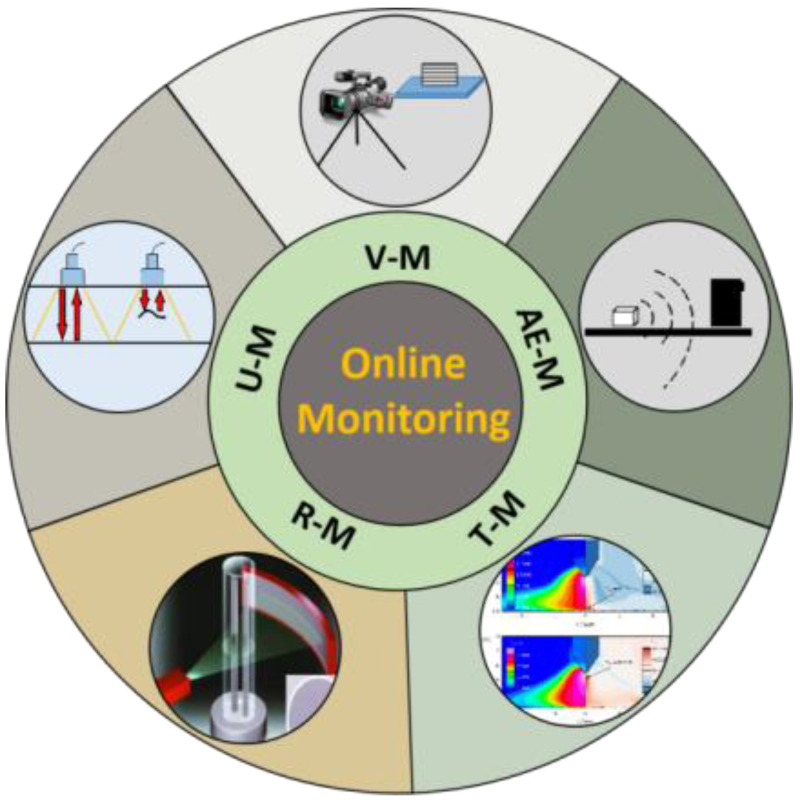
Schematic of commonly used in-process monitoring techniques for AM (V-M: Visual Monitoring; U-M: Ultrasonic Monitoring; R-M: Radiographic Monitoring; T-M: Thermographic Monitoring; AE-M: Acoustic Emission Monitoring).

**Figure 3 sensors-26-00438-f003:**
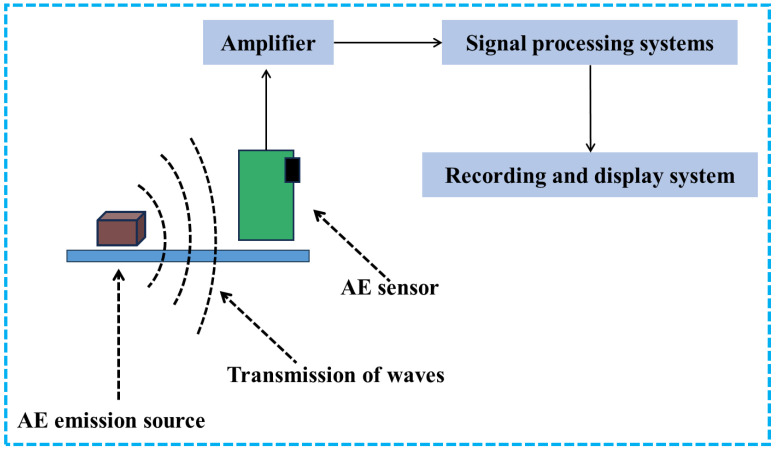
Schematic diagram of the acoustic emission principle.

**Figure 4 sensors-26-00438-f004:**
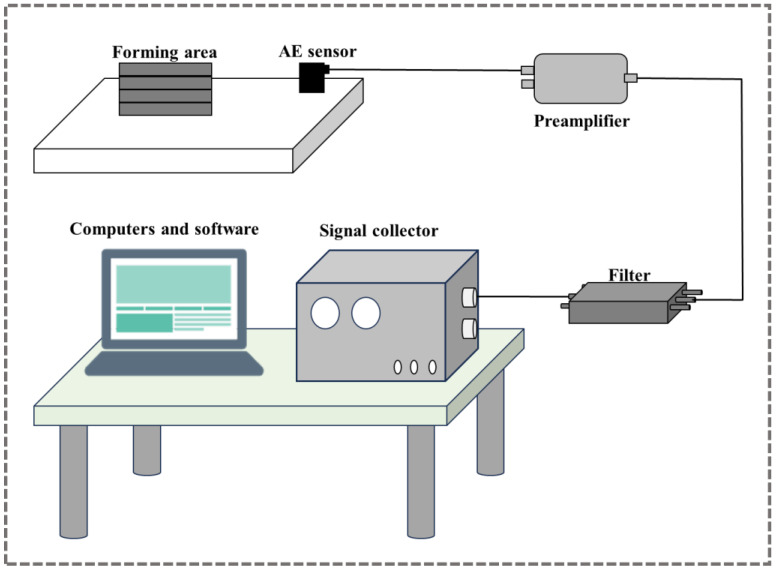
Schematic diagram of the acoustic emission system configuration.

**Figure 5 sensors-26-00438-f005:**
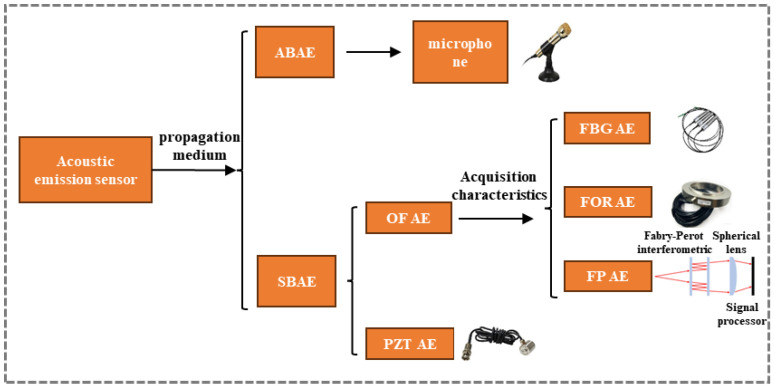
Classification of AE sensors.

**Figure 6 sensors-26-00438-f006:**
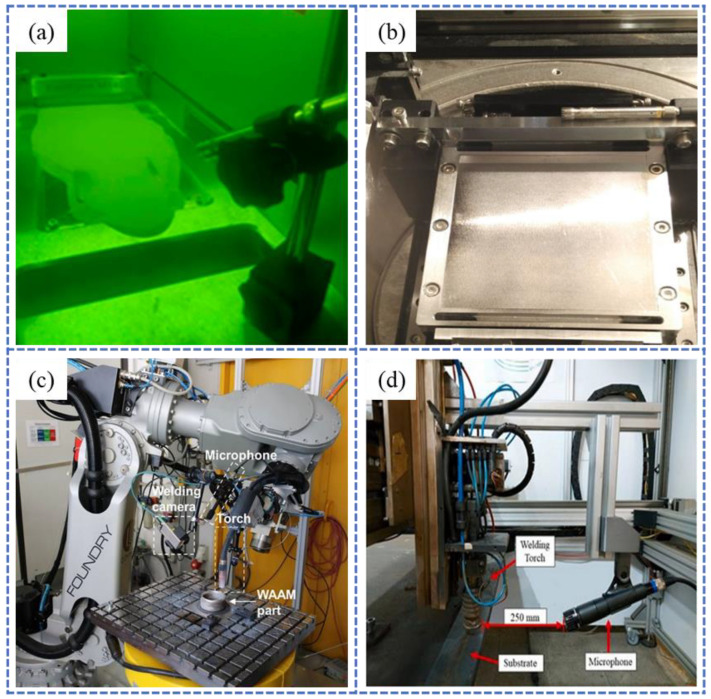
Typical microphone mounting configurations for AE monitoring: (**a**) 20 cm diagonally above the substrate in SLM [77]; (**b**) overhead beam mounting above the SLM substrate [80]; (**c**,**d**) 20 cm diagonally above the substrate in WA-DED [78,79].

**Figure 7 sensors-26-00438-f007:**
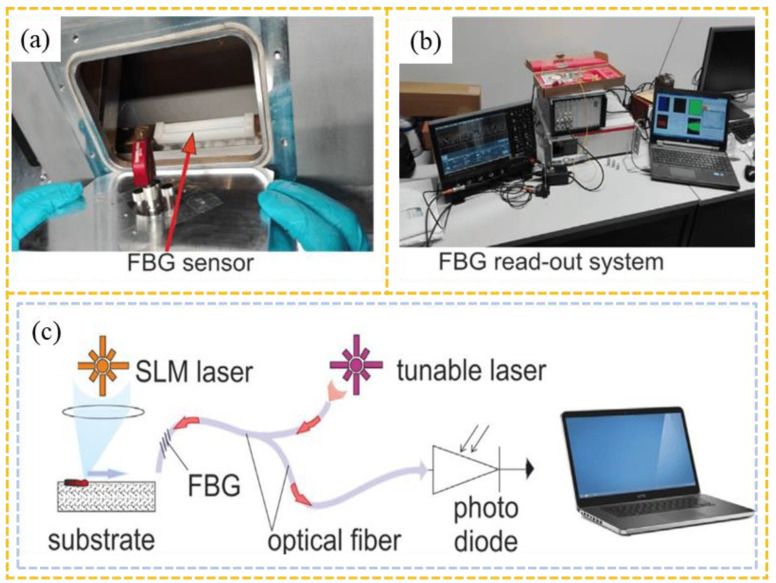
Configuration of FBG AE monitoring [81,82]: (**a**) Mounted on the sidewall of an SLM build chamber; (**b**) AE signal acquisition system; (**c**) Schematic diagram of the sensing principle.

**Figure 8 sensors-26-00438-f008:**
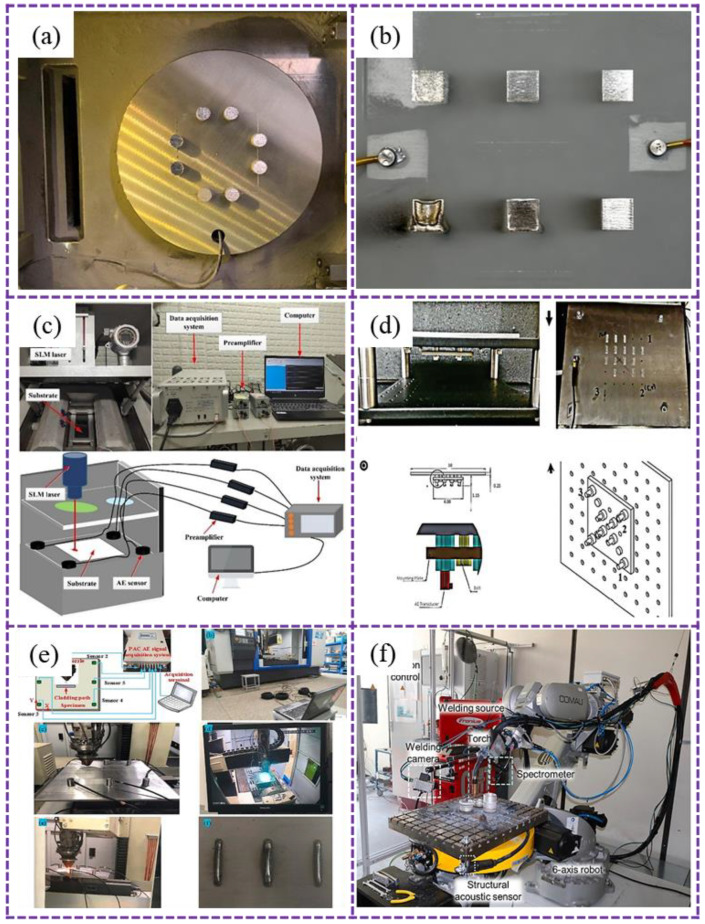
Mounting locations of PZT AE sensors: (**a**–**c**) on the substrate surface in SLM [37,83,84]; (**d**,**e**) on the substrate surface in L-DED [85,86,87]; (**f**) beneath the substrate platform in WA-DED [90].

**Figure 9 sensors-26-00438-f009:**
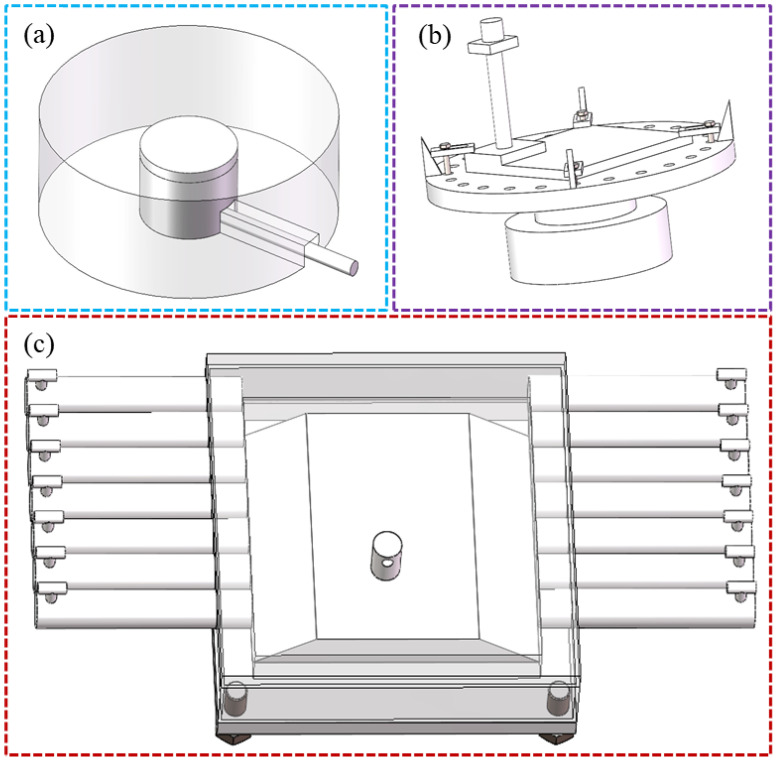
Schematic of the proposed solutions: (**a**) recessed substrate; (**b**) waveguide; (**c**) water-cooled substrate.

**Figure 10 sensors-26-00438-f010:**
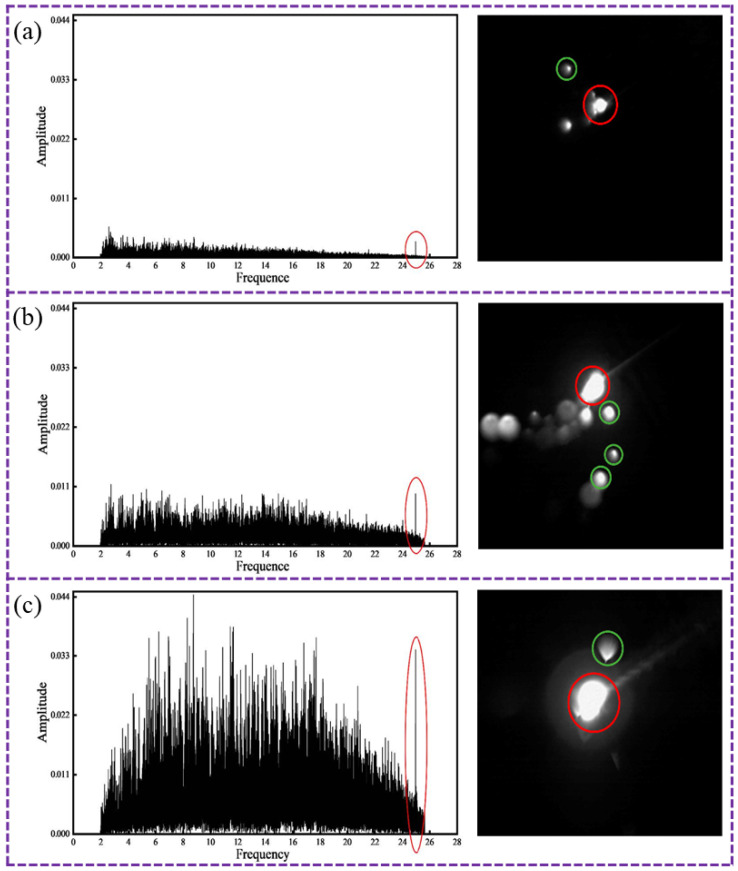
Frequency domain diagrams of molten pool sputtering images and sound signals under different laser powers [77]: (**a**) low laser power; (**b**) medium laser power; (**c**) high laser power.

**Figure 11 sensors-26-00438-f011:**
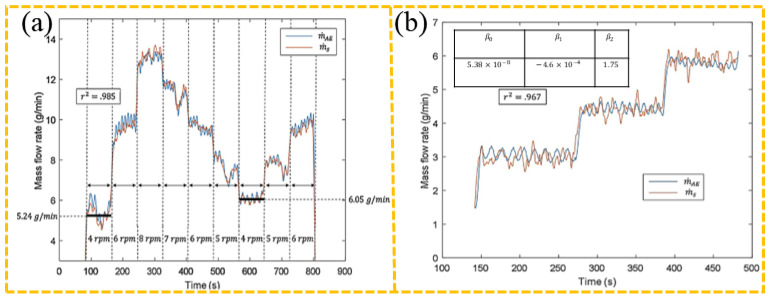
RMS value versus powder flow rate in L-DED, where $m_{AE}\;$represents the RMS of the AE signal and $m_{s}$ denotes the actual powder flow rate [92]: (**a**) the correlation between flow data and AE sensors; (**b**) AE system measures model parameters and mass flow.

**Figure 12 sensors-26-00438-f012:**
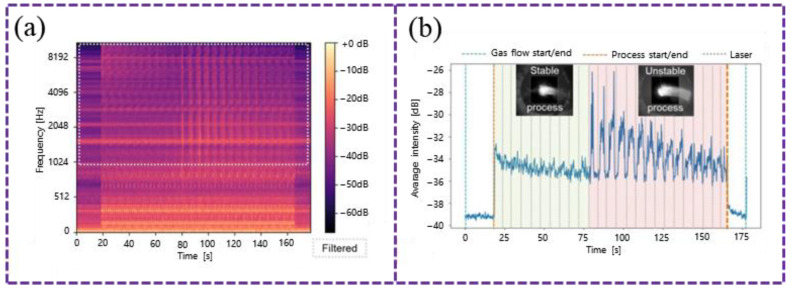
Characterization of process anomalies via the mean intensity of the Mel spectrum, showing the transition from stable to unstable conditions [78]: (**a**) mel spectrum; (**b**) mean intensity.

**Figure 13 sensors-26-00438-f013:**
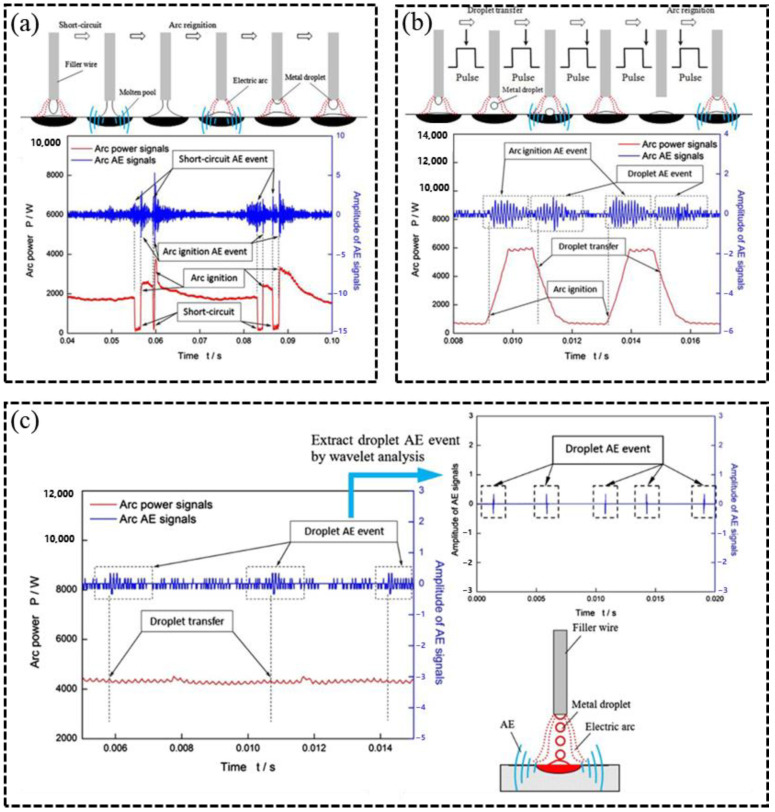
Time-domain correlation between AE signals and arc power under different transfer modes [88]: (**a**) non-pulsed CMT; (**b**) CMT+P; (**c**) projected droplet transfer.

**Figure 14 sensors-26-00438-f014:**
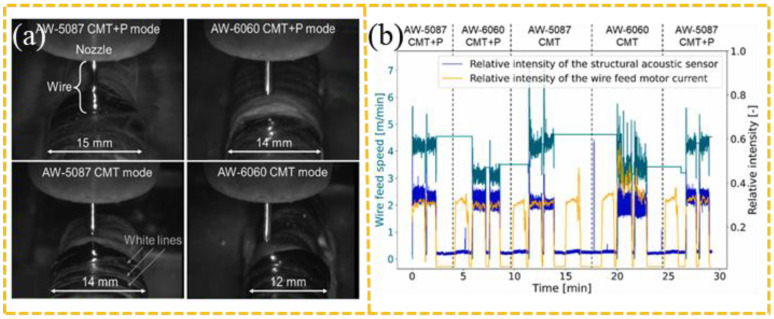
Under the CMT and CMT+P modes, the image signals and AE signals of AA5087 and AA6060 aluminum alloys [90]: (**a**) High-speed camera images; (**b**) graph showing the relationship between wire feeding speed and AE signal intensity.

**Figure 15 sensors-26-00438-f015:**
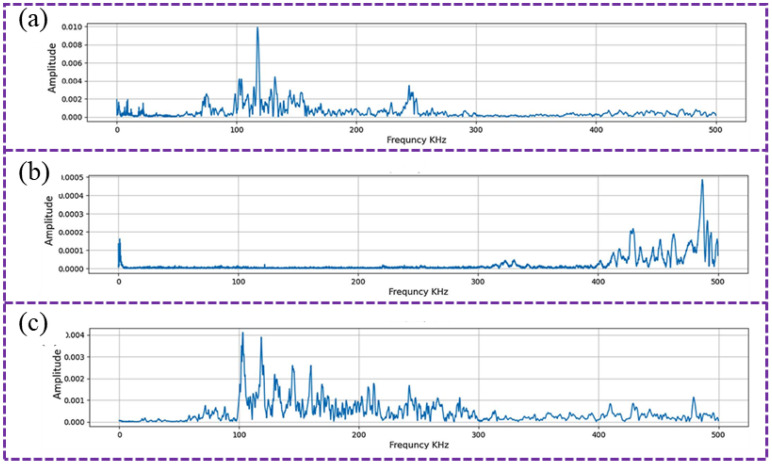
Spectral activity of AE signals corresponding to different defect types in SLM, with pore defects exhibiting higher activity in the high-frequency range [83]: (**a**) minimum defects; (**b**) crack; (**c**) porosity.

**Figure 16 sensors-26-00438-f016:**
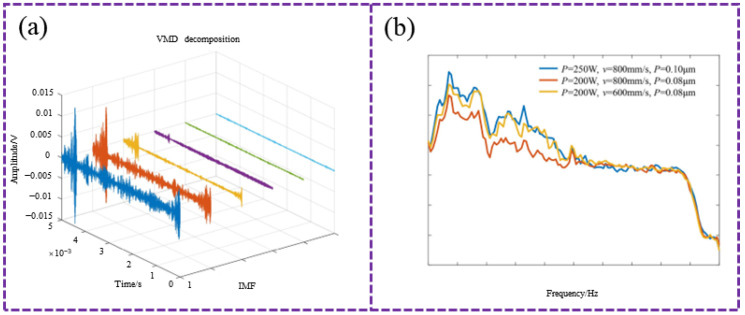
(**a**) Decomposition results of AE hit based on AEE-WOA-VMD method; (**b**) Power spectral density characteristics corresponding to different porosity levels (as reflected in parts of high, medium, and low quality) [84].

**Figure 17 sensors-26-00438-f017:**
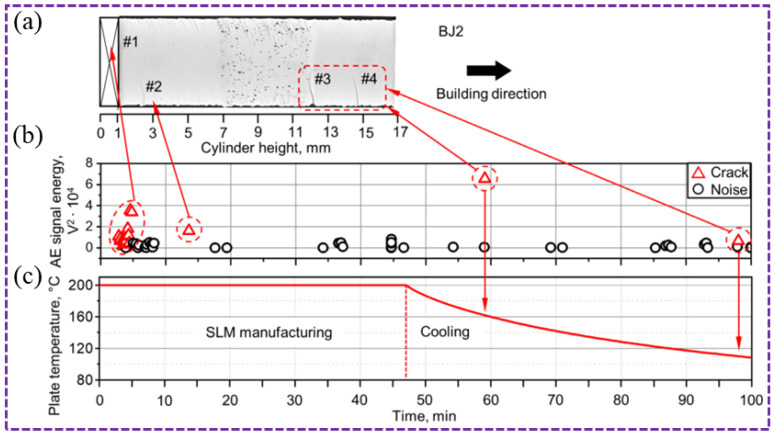
Correlation between AE energy and crack size in WA-DED (#1: highly scattered AE signals at the substrate-part interface; #2 and #4: low AE energy corresponding to small cracks; #3: high AE energy corresponding to a large crack) [91]: (**a**) cracks; (**b**) AE energy; (**c**) substrate temperature.

**Figure 18 sensors-26-00438-f018:**
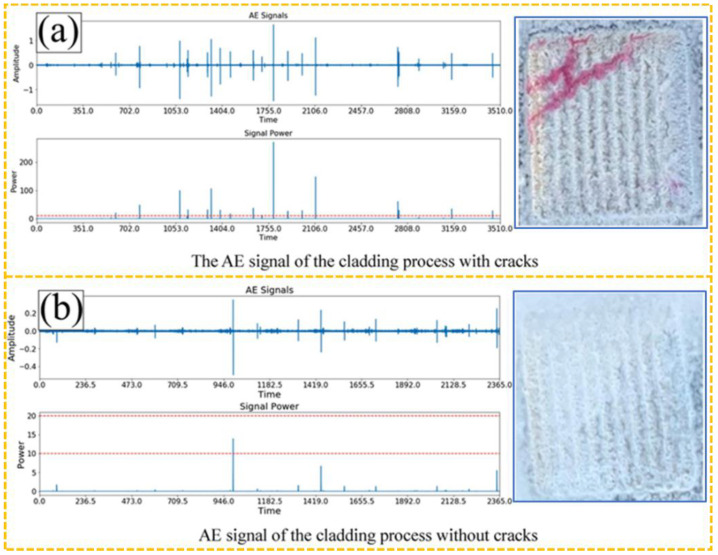
AE signals and real-time signal energy signals during the deposition process [122].

**Figure 19 sensors-26-00438-f019:**
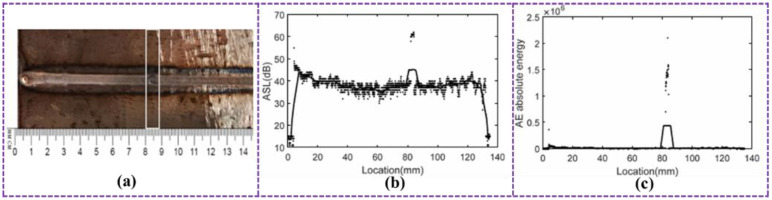
(**a**) Macroscopic morphology image of the sample; (**b**) ASL scanning image; (**c**) Ultrasonic absolute energy distribution map [126].

**Figure 20 sensors-26-00438-f020:**
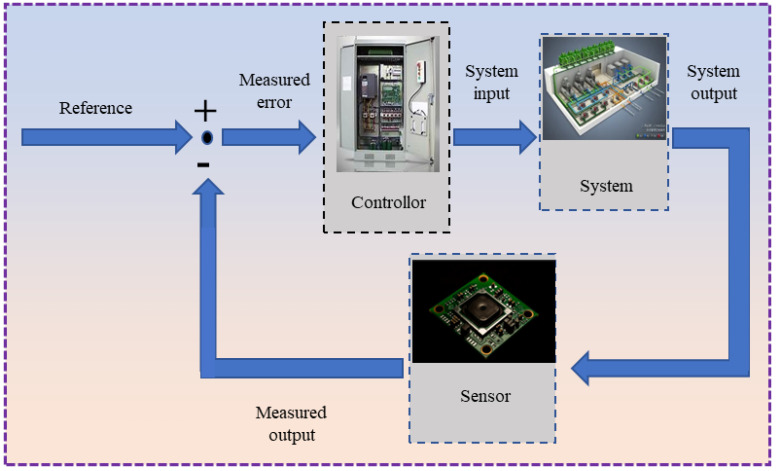
Schematic diagram of the feedback control principle.

**Figure 21 sensors-26-00438-f021:**
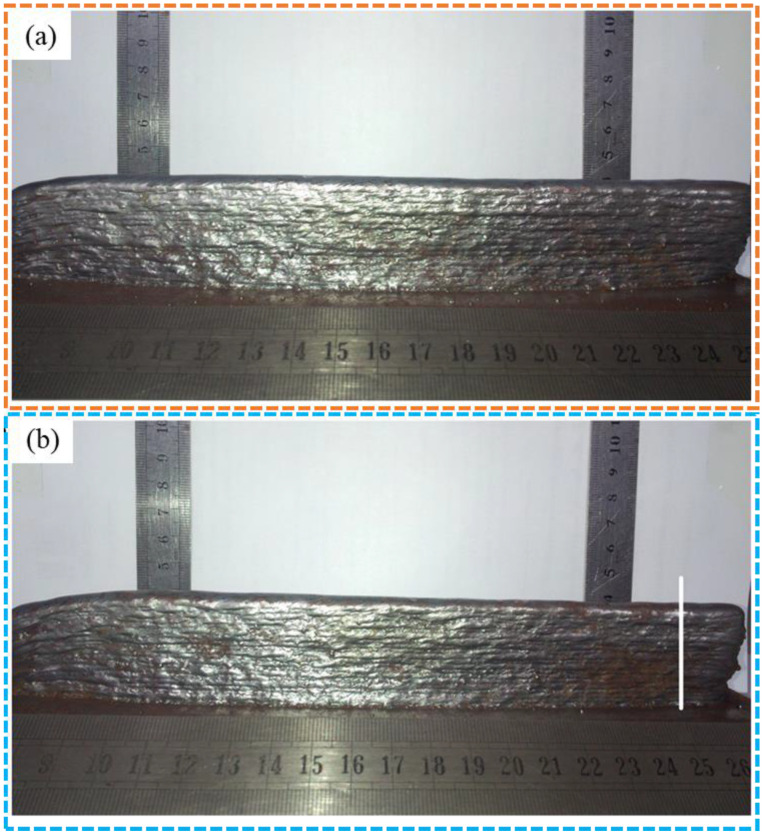
Closed-loop control of melt pool height using visual monitoring [147], demonstrating improved sidewall smoothness with feedback control: (**a**) open-loop control; (**b**) closed-loop control.

**Figure 22 sensors-26-00438-f022:**
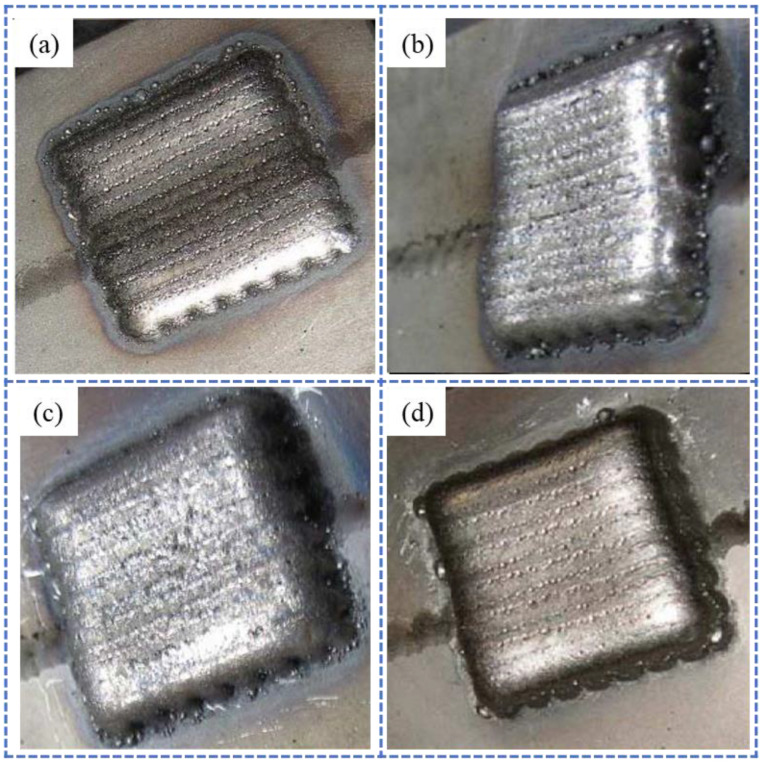
Thermal monitoring-based control of melt pool temperature [156], showing enhanced deposition fullness with increased thermal input compensation at higher layers: (**a**) 10th layer; (**b**) 20th layer; (**c**) 30th layer; (**d**) 40th layer.

**Table 1 sensors-26-00438-t001:** Analysis of the correlation between AE signal characteristics and defect types.

Types of Additive Manufacturing	Defect Characteristics	Method	Reference
LB-PBF	Crack and Porosity	Digital Noise Filter	[37]
	Porosity	Spectral convolutional neural networks (SCNN)	[81,82]
	Crack and Porosity	K-Means Clustering algorithm (K-Means) Gaussian Mixture Models (GMM) Variational auto encoders (VAE)	[83]
	Porosity	AEE-WOA-VMD	[84]
	Crack	Welch method	[91]
L-DED	Crack and Porosity	K-Means clustering (K-Means) Logistic Regression (LR), Artificial Neural Network (ANN)	[125]
WA-DED	Artificial Defects	Digital Noise Filter	[79]

**Table 2 sensors-26-00438-t002:** Monitoring models and their application results.

Modeling Approach	Applied to AE In-Process Monitoring of Metal AM	z	Outcome	Reference
Linear Regression	√	Powder flow rate in L-DED	95.9% accuracy for 316 L; 98.5% for CP-Ti	[92]
Fruit Fly Optimization	√	AE source localization in L-DED	Localization error: Layer 1: ±5.2 mm; Layer 2: ±5.8 mm; Layer 3: ±4.3 mm	[87]
K-Means Clustering	√	Identification of printing states in L-DED	>87% accuracy (low-freq); >70% accuracy (high-freq)	[86]
		Defect identification in SLM (pores/cracks)	90% accuracy	[83]
Neural Network	√	Spatter level classification in L-DED	85.08% accuracy	[77]
		Part quality prediction in SLM	83.3% accuracy	[84]
		Part quality classification in SLM	83% (high), 85% (medium), 89% (low) accuracy	[81]
		Porosity identification in SLM	78–91% accuracy	[82]
		Process state identification in L-DED	96.0% accuracy	[85]
Support Vector Machi	√	Part quality identification in SLM	91.7% accuracy	[84]
	×	Material extrusion status in FDM	95% accuracy	[141]
Random Forest	×	Fatigue crack growth pattern monitoring	97.6% accuracy	[142]

## Data Availability

All relevant data for reproducing the calculations are given in the main publication text.

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
