# Peer review of "A Review of In Situ Quality Monitoring in Additive Manufacturing Using Acoustic Emission Technology"

_sensors, 2026, doi:10.3390/s26020438_

Round 1

Reviewer 1 Report

Comments and Suggestions for Authors

Although acoustic emission has been used as a passive non-destructive evaluation method for monitoring crack growth in metals and composites, its ability to detect pores and defects during metal additive manufacturing represents a promising path for online process monitoring. This review presents an overview of major MAM techniques and current methods (including acoustic emission) used for process monitoring of MAM. A description of the major MAM techniques is provided, followed by an overview of common in-process monitoring methods (such as visual, ultrasonic, radiography, thermography, etc.). The focus of this review is on the use of the acoustic emission (AE) technique. A description of the technique is provided, followed by a detailed review of its applicability and effectiveness in monitoring process dynamics (such as spatter characteristics, melt pool dynamics and stability) and defects (pores, cracks) during manufacturing. An important discussion relating to "Intelligent Monitoring" is provided, which focuses on the use of artifical intelligence-based clustering algorithms to group specific AE characteristics to relevant MAM processes. Overall, this review article will be of interest to the MAM and AE research communities at large. I therefore recommend its publication in Sensors, with the following minor revisions: 

  1. Although the authors state that the choice of LBF or DED depends on the application, a couple of additional sentences explaining the primary differences and/or advantages/ drawbacks between PBF and DED methods for metal additive manufacturing will be beneficial in the Introduction section.
  2. Please fix the errors in the labels in Figure 1. PBF-LB has been labeled twice - and PBF-EB is missing.

  3. Besides conventional ultrasound for online process monitoring of metal additive manufacturing, the use of laser-generated ultrasound using all-optical ultrasound excitationa and detection for MAM process monitoring should also be discussed. A recent study includes the 2022 paper from Lawrence Livermore National Laboratory: 

    https://www.nature.com/articles/s41598-022-07261-w

  4.  

    Please clarify how Fiber Bragg Grating sensors work - do they detect changes in the refractive index of air induced by high frequency sonic/ ultrasonic waves generated by the build process?

  5. An explanation as to why the detected AE frequency can correlate with spatter behavior is needed. Is it related to the energy of the impact of the molten metal on the base plate? A physics-informed reason for this is needed and helpful for the readers.

  6. What is AEE-WOA-VMD? This is mentioned without defining the acrnoym or providing a context.

  7. In Table 1, it is good to see the accuracy levels summarized for each AE modeling approach for the stated application. A discussion on the lower accuracy examples is needed. What are the potential reasons for lower accuracy?
  8. In the Future Perspectives section (final paragraphs), more details on the authors' opinions are needed. What is the authors' opinion of intelligent additive manufacturing? Will it involve an integrated approach utilizing multiple NDE process monitoring tools? How does AE fill in the deficiencies of existing methods or how does it compliment other techniques. I believe the final section in its current form lacks detailed perspective.

Author Response

Response to Reviewer 1 Comments

Comments 1: Although the authors state that the choice of LBF or DED depends on the application, a couple of additional sentences explaining the primary differences and/or advantages/ drawbacks between PBF and DED methods for metal additive manufacturing will be beneficial in the Introduction section.

Response 1: Thank you for pointing this out. We agree with this comment. Therefore, we have made the necessary revisions.

“The PBF technology selectively melts the pre-prepared powder on the powder bed, excelling in manufacturing high-precision and complex-shaped end parts with excellent density and surface quality. However, its forming size is limited and the cost is relatively high. The DED technology directly feeds the powder or wire into the molten pool for deposition, featuring a high printing rate, the ability to repair large-sized parts, good flexibility, but with lower forming accuracy and requiring a large amount of subsequent processing.” (Please see to Section 1.1, Paragraph 2, lines 4 to 10. These contents have been modified using red font.)

Comments 2: Please fix the errors in the labels in Figure 1. PBF-LB has been labeled twice - and PBF-EB is missing.

Response 2: Thank you for pointing this out. We agree with this comment. Therefore, we have made the necessary revisions. (Please see to the content Fig. 1 in the manuscript, and note that the manuscript has been updated.)

Fig. 1. Classification of Metal Additive Manufacturing Technologies.

Comments 3: Besides conventional ultrasound for online process monitoring of metal additive manufacturing, the use of laser-generated ultrasound using all-optical ultrasound excitationa and detection for MAM process monitoring should also be discussed. A recent study includes the 2022 paper from Lawrence Livermore National Laboratory:

 https://www.nature.com/articles/s41598-022-07261-w.

Response 3: Thank you for pointing this out. We agree with this comment. Therefore, we have made the necessary revisions.

However, Harke et al. [54] also conducted research on the relationship between the surface acoustic wave (SAW) signals generated by laser ultrasonic waves and the surface and subsurface defects during the L-PBF forming process. (Please see to Section 1.2, Paragraph 7, last 3 lines. These contents have been modified using red font.)

Comments 4: Please clarify how Fiber Bragg Grating sensors work - do they detect changes in the refractive index of air induced by high frequency sonic/ ultrasonic waves generated by the build process?

Response 4: Thank you for pointing this out. We agree with this comment. Therefore, we have made the necessary revisions. The primary detection mechanism of the Fiber Bragg Grating (FBG) sensors is not based on changes in the refractive index of the surrounding air. Instead, the high-frequency sonic/ultrasonic waves generated by the build process physically impinge on the optical fiber, inducing dynamic strain within the fiber itself. This strain directly alters both the period of the grating  and the effective refractive index ( ) of the fiber core through the photo-elastic effect, resulting in a measurable shift of the Bragg wavelength . The air serves as the propagation medium for the acoustic energy, but the FBG fundamentally responds to the resulting mechanical deformation. We have revised the manuscript to include this clarification.

“The primary detection mechanism of the FBG sensors is not based on changes in the refractive index of the surrounding air. Instead, the high-frequency sonic/ultrasonic waves generated by the build process physically impinge on the optical fiber, inducing dynamic strain within the fiber itself. This strain directly alters both the period of the grating  and the effective refractive index  of the fiber core through the photo-elastic effect, resulting in a measurable shift of the Bragg wavelength . The air serves as the propagation medium for the acoustic energy, but the FBG fundamentally responds to the resulting mechanical deformation.” (Please see to Section 2.1, Paragraph 3, last 8 lines. These contents have been modified using red font.)

Comments 5: An explanation as to why the detected AE frequency can correlate with spatter behavior is needed. Is it related to the energy of the impact of the molten metal on the base plate? A physics-informed reason for this is needed and helpful for the readers.

Response 5: Thank you for pointing this out. We agree with this comment. Therefore, we have made the necessary revisions.

“During the melting process, the acoustic emission equipment monitors the acoustic signals of the melted and non-melted materials in the time domain and frequency domain. Through time-domain and frequency-domain analysis, noise can be removed and the frequency range of the melting acoustic signals can be determined. At the same time, different agglomeration and rupture phenomena generated under different melting conditions are analyzed [80].” (Please see to Section 3.2.1, Paragraph 6, last 8 lines. These contents have been modified using red font.)

Comments 6: What is AEE-WOA-VMD? This is mentioned without defining the acrnoym or providing a context.

Response 6: Thank you for pointing this out. We agree with this comment. Therefore, we have made the necessary revisions. AEE-WOA-VMD is a key algorithm introduced in our work, which integrates Adaptive Ensemble Entropy (AEE) as a fitness function to guide the Whale Optimization Algorithm (WOA) for the optimal parameter selection of Variational Mode Decomposition (VMD). We have revised the manuscript to explicitly define this acronym and provide a brief explanation where it is first introduced. We appreciate the comment, which has helped improve the clarity of our paper.

“After processing the AE signals with Adaptive Ensemble Entropy - Whale Optimization Algorithm - Variational Mode Decomposition (AEE-WOA-VMD),   .“ (Please see to Section 3.2.2, Paragraph 3, lines 3 to 5.)

Comments 7: In Table 1, it is good to see the accuracy levels summarized for each AE modeling approach for the stated application. A discussion on the lower accuracy examples is needed. What are the potential reasons for lower accuracy?

Response 7: Thank you for pointing this out. We agree with this comment. Therefore, we have made the necessary revisions. In response, we have added a dedicated discussion in Section 4.2 of the revised manuscript to analyze the potential reasons for the lower accuracy observed in some of the AE modeling approaches summarized in Table 1. The performance variations are attributed to a combination of factors, primarily including the inherent complexity and spectral overlap of AE signatures from certain physical phenomena, which challenge the feature extraction capability of simpler models. Furthermore, the performance of some models was constrained by the limited quantity of high-quality training data available for specific, rare defect types, leading to insufficient feature learning. The suboptimal performance of conventional machine learning models can also be linked to their dependence on hand-crafted features and their limited capacity to capture the complex, non-linear patterns in raw waveform data compared to deep learning architectures. This discussion provides a more critical perspective on the results and underscores the context-dependent nature of model selection for AE analysis.

“The performance variations are attributed to a combination of factors, primarily including the inherent complexity and spectral overlap of AE signatures from certain physical phenomena, which challenge the feature extraction capability of simpler models.  Furthermore, the performance of some models was constrained by the limited quantity of high-quality training data available for specific, rare defect types, leading to insufficient feature learning. The suboptimal performance of conventional machine learning models can also be linked to their dependence on hand-crafted features and their limited capacity to capture the complex, non-linear patterns in raw waveform data compared to deep learning architectures.” (Please see in Section 3.3.1.6, Paragraph 3.)

Comments 8: In the Future Perspectives section (final paragraphs), more details on the authors' opinions are needed. What is the authors' opinion of intelligent additive manufacturing? Will it involve an integrated approach utilizing multiple NDE process monitoring tools? How does AE fill in the deficiencies of existing methods or how does it compliment other techniques. I believe the final section in its current form lacks detailed perspective.

Response 8: Thank you for pointing this out. We agree with this comment. Therefore, we have made the necessary revisions. We believe that the development of intelligent additive manufacturing will inevitably lead to an integrated monitoring system that combines multiple sources of information. A single monitoring technology is insufficient to comprehensively capture the complex physical phenomena (such as the coupling effects of heat, force, and metallurgy) during the additive manufacturing process. Therefore, the future intelligent monitoring platform will no longer be a simple combination of multiple non-destructive testing technologies, but will achieve real-time synchronous collection and intelligent fusion analysis of various sensing data through a digital twin framework or edge computing platform. This integrated approach aims to achieve a closed-loop intelligent manufacturing process from “monitoring” to “prediction” and “control”, that is, when an abnormality is detected in the process, it can adjust the process parameters (such as laser power, scanning speed) in real time, thereby actively suppressing the occurrence of defects instead of conducting post-event detection.

“As relevant technologies continue to advance, additive manufacturing-distinct from conventional subtractive approaches-is poised to become a dominant paradigm for component fabrication. Developing AE-based in-process monitoring for AM processes thus represents a highly valuable endeavor. Future efforts should focus on establishing robust in-process monitoring capabilities coupled with closed-loop control mechanisms, ultimately realizing an integrated “process-monitoring-feedback adjustment” framework for intelligent additive manufacturing. This framework integrates multiple non-destructive testing technologies into a closed-loop intelligent system, enabling multi-scale and all-round perception from macroscopic process stability to microscopic structural integrity. This multi-technology integration strategy will provide a solid data foundation for building an intelligent additive manufacturing system with true self-sensing, self-diagnosis, and self-repair capabilities.” (Please see in Section 4, Paragraph 2.)

Reviewer 2 Report

Comments and Suggestions for Authors

This manuscript presents a timely and comprehensive review on the application of acoustic emission (AE) technology for in-process monitoring of metal additive manufacturing (MAM). The topic is of significant interest to the field, addressing a critical need for reliable quality assurance in MAM processes. The review is well-structured and provides a thorough overview of the fundamental principles of AE, hardware configurations, correlations between AE features and process anomalies/defects, and the integration of intelligent monitoring models. The authors have compiled an extensive and relevant body of literature to support their discussion. However, to further enhance the impact and clarity of this otherwise valuable contribution, the manuscript would benefit from a more critical and forward-looking analysis in several areas. The following revisions are required:

  1. The article title may be revised to refine it without altering its meaning.
  2. All images in the text are of poor quality; all images in the text require modification. All image captions in the text must be provided in the manuscript and analyz
  3. Numerous punctuation errors are present throughout the manuscript, such as in the introductory section where “bottom-up” and “3D Printing” are incorrectly placed. Please review the entire text and make the necessary corrections.
  4. In the introduction, please provide professional descriptions for “Laser Beam (DED-LB/LENS/LCF), Electron Beam (DED-EB), and Arc (DED-ARC/WAAM)”, ensuring consistency throughout the text and making the necessary revisions.
  5. The overall structure is logically coherent, though the transition between Sections 3.1 and 3.2 could be smoother. Consider adding a brief introductory sentence linking the hardware configuration to the parameter characteriz
  6. The paper is generally well written, but there are instances of inconsistent terminology (e.g., “spatter” and “splatter”). Please ensure that terminology is used consistently throughout the text.
  7. The reference list is comprehensive and relevant. However, it lacks some of the most recent high-impact papers on AE in additive manufacturing (e.g., those published between 2023 and 2025), which should be incorporated to ensure the review content remains up to date.

Author Response

Response to Reviewer 2 Comments

Comments 1: The article title may be revised to refine it without altering its meaning.

Response 1: Thank you for pointing this out. We agree with this comment. Therefore, we have made the necessary revisions.

“A Review of In-Situ Quality Monitoring in Additive Manufacturing Using Acoustic Emission Technology” (Please look at the title on the front page of the article, which is marked in blue font.)

Comments 2: All images in the text are of poor quality; all images in the text require modification. All image captions in the text must be provided in the manuscript and analyz.

Response 2: Thank you for pointing this out. We agree with this comment. Therefore, we have made the necessary revisions.

“Metal additive manufacturing (MAM), an important branch in the field of AM, encompasses a variety of specific processing methods. Its classification is shown in Fig. 1.” (Please see to Section 1.1, Paragraph 2, lines 1 to 3. These contents have been modified using blue font.)

Fig. 1. Classification of MAM Technologies.

“At present, there are various technologies mainly practiced for online monitoring in the field of AM, as shown in Fig. 2, including visual monitoring, ultrasonic monitoring, radiographic monitoring, thermal imaging monitoring, and acoustic emission monitoring, etc.” (Please see to Section 1.2, Paragraph 2. These contents have been modified using blue font.)

Fig. 2. Schematic of commonly used in-process monitoring techniques for AM (V-M: Visual Monitoring; U-M: Ultrasonic Monitoring; R-M: Radiographic Monitoring; T-M: Thermographic Monitoring; AE-M: Acoustic Emission Monitoring).

“The basic composition of a sound emission monitoring system is shown in Fig. 3.” (Please see to Section 2, the first line of the second paragraph. These contents have been modified using blue font.)

Fig. 3. Schematic diagram of the acoustic emission principle.

“The composition of the acoustic emission system is shown in Fig. 4, …” (Please see to Section 2.1, Paragraph 1, lines 1 to 3. These contents have been modified using blue font.)

Fig. 4. Schematic diagram of the acoustic emission system configuration.

“The classification of AE sensors is shown in Fig. 5.” (Please see to Section 2.1, Paragraph 2, lines 3 to 4. These contents have been modified using blue font.)

Fig. 5. Classification of AE sensors.

“The microphone uses air as the transmission medium and does not need to come into direct contact with the work area. Common installation positions are shown in Fig. 6.” (Please see to Section 3.1, Paragraph 2, lines 1 to 3. These contents have been modified using blue font.)

Fig. 6. Typical microphone mounting configurations for AE monitoring: (a) 20 cm diagonally above the substrate in SLM; (b) overhead beam mounting above the SLM substrate;

(c, d) 20 cm diagonally above the substrate in WA-DED.

“For instance, …, as shown in Fig. 7.” (Please see to Section 3.1, Paragraph 2, lines 9 to 10. These contents have been modified using blue font.)

Fig. 7. Configuration of FBG AE monitoring: (a) Mounted on the sidewall of an SLM build chamber; (b) AE signal acquisition system; (c) Schematic diagram of the sensing principle.

“The common installation positions of PZT sensors are shown in Fig. 8.” (Please see to Section 3.1, Paragraph 4, lines 1 to 2. These contents have been modified using blue font.)

Fig. 8. Mounting locations of PZT AE sensors: (a-c) on the substrate surface in SLM; (d-e) on the substrate surface in L-DED; (g) beneath the substrate platform in WA-DED.

“For PBF systems, a recess can be machined beneath the build platform to accommodate a PZT sensor, as shown in Fig. 9(a), accompanied by a dedicated channel for cable routing.” (Please see to Section 3.1, Paragraph 5, lines 2 to 4. These contents have been modified using blue font.)

“For DED systems, particularly in WA-DED, a waveguide fabricated from a low-acoustic-attenuation material can be welded to the substrate, as shown in Fig. 9(b).” (Please see to Section 3.1, Paragraph 5, lines 5 to 7. These contents have been modified using blue font.)

“Alternatively, an integrated water-cooling system can be applied to the substrate, as shown in Fig. 9(c).” (Please see to Section 3.1, Paragraph 5, lines 7 to 8. These contents have been modified using blue font.)

Fig. 9. Schematic of the proposed solutions: (a) recessed substrate;

(b) waveguide; (c) water-cooled substrate.

“Whiting et al. detected powder flow velocity in L-DED by analyzing AE signals generated from powder impacts on the feed channel, showing that the root mean square (RMS) value serves as a reliable indicator, as shown in Fig. 10(a).” (Please see to Section 3.2.1, Paragraph 2. These contents have been modified using blue font.)

Fig. 10. Correlation analysis of AE parameters with process variables: (a) RMS value versus powder flow rate in L-DED, where represents the RMS of the AE signal and  denotes the actual powder flow rate; (b) AE signal intensity as a function of wire feed speed for AA5087 and AA6060 aluminum alloys under CMT and CMT+P modes.

“By analyzing the periodicity of these peaks, different WA-DED deposition modes can be effectively identified, as shown in Fig. 11.” (Please see to Section 3.2.1, Paragraph 4, line 4. These contents have been modified using blue font.)

Fig. 11. Time-domain correlation between AE signals and arc power under different transfer modes: (a) non-pulsed CMT; (b) CMT+P; (c) projected droplet transfer.

“They identified a peak at 25 kHz that effectively distinguished different levels of spatter, as shown in Fig. 12.” (Please see to Section 3.2.1, Paragraph 7, lines 7 to 9. These contents have been modified using blue font.)

Fig. 12. Frequency domain diagrams of molten pool sputtering images and sound signals under different laser powers: (a) low laser power; (b) medium laser power; (c) high laser power.

“……, while Tobias et al. found that abrupt changes in the average intensity of the Mel spectrum can reliably indicate sudden anomalies during otherwise stable L-DED operation, as shown in Fig. 13.” (Please see to Section 3.2.1, Paragraph 8, lines 9 to 11. These contents have been modified using blue font.)

Fig. 13. Characterization of process anomalies via the mean intensity of the Mel spectrum, showing the transition from stable to unstable conditions: (a) mel spectrum; (b) mean intensity.

“After processing the AE signals with AEE-WOA-VMD, as shown in Fig. 14(a), they found that power spectral density and energy entropy effectively distinguished different porosity levels, as shown in Fig. 14(b).”(Please see to Section 3.2.2, Paragraph 3, line 5. These contents have been modified using red and blue font.)

Fig. 14. (a)Decomposition results of AE hit based on AEE-WOA-VMD method; (b)Power spectral density characteristics corresponding to different porosity levels (as reflected in parts of high, medium, and low quality).

“Mohammad et al. demonstrated that the spectral distribution of AE signals can differentiate between cracks and porosity by fabricating specimens with controlled defects through variation of SLM laser power, as shown in Fig. 15.” (Please see to Section 3.2.2, Paragraph 5, lines 13 to 15. These contents have been modified using blue font.)

Fig. 15. Spectral activity of AE signals corresponding to different defect types in SLM, with pore defects exhibiting higher activity in the high-frequency range:

(a)    defect-free; (b) porosity; (c) crack.

“By correlating AE signals with micro-computed tomography, Mikhail et al. showed that time-domain features such as RMS, standard deviation, and peak amplitude can characterize micro-crack formation during SLM fabrication and cooling, with AE energy levels even enabling crack size differentiation, as shown in Fig. 16.” (Please see to Section 3.2.2, Paragraph 6, lines 2 to 6. These contents have been modified using blue font.)

Fig. 16. Correlation between AE energy and crack size in WA-DED (#1: highly scattered AE signals at the substrate-part interface; #2 and #4: low AE energy corresponding to small cracks; #3: high AE energy corresponding to a large crack): (a) cracks; (b) AE energy; (c) substrate temperature.

“In the context of AE monitoring, this approach enables real-time process adjustment by continuously analyzing AE signals and dynamically refining process parameters to maintain optimal manufacturing conditions and ensure consistent product quality, as shown in Fig. 17.” (Please see to Section 3.3.2, Paragraph 1, lines 5 to 9. These contents have been modified using blue font.)

Fig. 17. Schematic diagram of the feedback control principle.

“For instance, Jun et al. developed a control algorithm to automatically adjust the nozzle-to-substrate distance (NTSD) in GMAW, effectively regulating the melt pool height, as shown in Fig. 18.” (Please see to Section 3.3.2, Paragraph 2, lines 6 to 8. These contents have been modified using blue font.)

Fig. 18. Closed-loop control of melt pool height using visual monitoring, demonstrating improved sidewall smoothness with feedback control: (a) open-loop control; (b) closed-loop control.

“Song et al., used a dual-color high-temperature meter to measure the molten pool temperature during laser cladding. The dynamic relationship between laser power and the molten pool was described using a state-space model. The temperature was experimentally identified using subspace methods. The closed-loop process can track the molten pool temperature to the reference temperature curve by adjusting the diode to control the laser power. The compensatory deficiency was verified using cladding for GPC. as shown in Fig. 19.” (Please see to Section 3.3.2, Paragraph 3, lines 8 to 9. These contents have been modified using blue font.)

Fig. 19. Thermal monitoring-based control of melt pool temperature, showing enhanced deposition fullness with increased thermal input compensation at higher layers: (a) 10th layer;

(b)    20th layer; (c) 30th layer; (d) 40th layer.

In addition, we have also carefully revised the entire text and marked it in blue.

Comments 3: Numerous punctuation errors are present throughout the manuscript, such as in the introductory section where “bottom-up” and “3D Printing” are incorrectly placed. Please review the entire text and make the necessary corrections.

Response 3: Thank you for pointing this out. We agree with this comment. Therefore, we have made the necessary revisions.

“Additive Manufacturing (AM) is a process that constructs three-dimensional objects layer-by-layer from a digital model, following a “bottom-up” approach contrary to traditional subtractive methods. This technology, also known as “3D Printing”, “Rapid Prototyping”, or “Solid Freeform Fabrication” depending on specific process characteristics” (Please see to Section 1.1, Paragraph 1, lines 2 to 4. These contents have been modified using blue font.)

Comments 4: In the introduction, please provide professional descriptions for “Laser Beam (DED-LB/LENS/LCF), Electron Beam (DED-EB), and Arc (DED-ARC/WAAM)”, ensuring consistency throughout the text and making the necessary revisions.

Response 4: Thank you for pointing this out. We agree with this comment. Therefore, we have made the necessary revisions.

“PBF techniques are further classified according to the energy source employed, namely laser beam powder bed fusion (LB-PBF) and electron beam powder bed fusion (EB-PBF). The LB-PBF process can be subdivided based on laser power and the extent of material melting into selective laser sintering (SLS) and selective laser melting (SLM). Conversely, DED techniques utilize a wider range of energy sources, including laser directed energy deposition (L-DED), electron beam directed energy deposition (EB-DED), and wire arc directed energy deposition (WA-DED).” (Please see to Section 1.1, Paragraph 2, lines 11 to 17. These contents have been modified using blue font.)

Comments 5: The overall structure is logically coherent, though the transition between Sections 3.1 and 3.2 could be smoother. Consider adding a brief introductory sentence linking the hardware configuration to the parameter characterize.

Response 5: Thank you for pointing this out. We agree with this comment. Therefore, we have made the necessary revisions.

“Effective signal denoising is a necessary condition for accurately extracting feature parameters from the data.” (Please see to Section 3.2, Paragraph 1, line 1. These contents have been modified using blue font.)

Comments 6: The paper is generally well written, but there are instances of inconsistent terminology (e.g., “spatter” and “splatter”). Please ensure that terminology is used consistently throughout the text.

Response 6: Thank you for pointing this out. We agree with this comment. Therefore, we have made the necessary revisions.

Shu et al. [77] employed a convolutional neural network (CNN) with acoustic signals as the input layer and spatter images as the output, performing dimensional reduction through feature extraction to classify and predict different levels of in L-DED, achieving accuracy above 85.08%.” (Please see to Section 3.3.1.4, Paragraph 2, lines 3 to 6. These contents have been modified using blue font.)

Comments 7: The reference list is comprehensive and relevant. However, it lacks some of the most recent high-impact papers on AE in additive manufacturing (e.g., those published between 2023 and 2025), which should be incorporated to ensure the review content remains up to date.

Response 7: Thank you for pointing this out. We agree with this comment. Therefore, we have made the necessary revisions.

“The AE online monitoring technology analyzes the sound wave signals generated during the material processing to provide an efficient monitoring method for the AM process. This technology does not require integrating sensors into the processing head, and it is flexible in deployment, enabling low-cost and real-time online process monitoring [50]. In practical applications, Raeker et al. [51] utilized piezoelectric transduce (PZT) sensors and AE technology to achieve in-situ crack detection in a single-channel laser melting experiment. By analyzing the acoustic emission signals, they revealed the influence patterns of different laser powers and scanning speeds on crack size, as well as the crack initiation characteristics of the material. Ansari et al. [52] proposed a monitoring method based on the exponential decay characteristics of AE signals. Through the analysis of the second derivative, noise was effectively filtered out, enabling the identification, quantification of surface and internal cracks during DED, as well as the reliable correlation of their initiation time and location. Furthermore, Xu et al. [53] combined AE technology with machine learning algorithms to systematically analyze the influence of key process parameters such as overlap rate, dwell time, and layer number on the relationship between AE features and Rockwell hardness. By integrating strain sensors, they achieved real-time and non-destructive prediction of the hardness of L-DED components, demonstrating a promising application in engineering.”

(Please see to Section 1.2, Paragraph 7. These contents have been modified using blue font.)

References:

[50] M.J. Ansari, E.J.G. Arcondoulis, A. Roccisano, C. Schulz, T. Schlaefer, C. Hall, Optimized analytical approach for the detection of process-induced defects using acoustic emission during directed energy deposition process, Additive Manufacturing 86 (2024) 104218.

[51] E.B. Raeker, N. Tulshibagwale, K.M. Mullin, J.D. Lamb, T.M. Pollock, Detection and Classification of Cracking Via Acoustic Emission During Laser-Melting Screening for Additive Manufacturing, JOM 77(10) (2025) 7274-7286.

[52] M.J. Ansari, A. Roccisano, E.J.G. Arcondoulis, C. Schulz, T. Schläfer, C. Hall, Relationship between associated acoustic emission and crack position during directed energy deposition of a metal matrix composite, Journal of Manufacturing Processes 147 (2025) 177-190.

[53] K. Xu, Y. Mahmoud, S. Manoochehri, C.K.P. Vallabh, Using acoustic emission signal analysis and machine learning algorithms for predicting mechanical hardness in laser directed energy deposition parts, INTERNATIONAL JOURNAL OF ADVANCED MANUFACTURING TECHNOLOGY 138(9-10) (2025) 4455-4474.

Reviewer 3 Report

Comments and Suggestions for Authors

The authors did a commendable job with presenting a comprehensive review of acoustic emission (AE) technology in metal additive manufacturing (MAM), covering topics from fundamental principles to advanced applications. A broad overview of MAM processes and in-process monitoring methods were provided, before delving into AE sensor configurations, signal feature extraction techniques, and the use of AE for real-time monitoring of process anomalies and internal defects. Overall, the paper’s major strengths are, logical organization, and the attempt to not only summarize existing work but also propose solutions and future research avenues for improving AE-based monitoring in additive manufacturing.

The authors are requested to address a few comments that will make the manuscript more regourous, up-to-date, and help this manuscript to be accepted for publication. 

  1. The authors are encouraged to update the manuscript with state-of-the-art literature in this filed. Especially since this is a literature review paper, it is quentessential for the authors to cite more recent papers in this field of research. Only 2 papers from 2025 and 2 from 2024 are cited. A few publications pertinent to this research are provided in the comments to follow.

  2. While classifying metal additive manufacturing technologies, the authors missed out an another important classification - "Binder-based AM techniques".
    This class of MAM technologies encompass binder jetting, material extrusion (MEX), and material jetting. Please expand on these technologies in your write-up. A few references to these technologies are provided below.

  3. References to Binder jetting:
    1. https://doi.org/10.1108/RPJ-12-2014-0180
    2. https://doi.org/10.1016/j.promfg.2017.07.084
    3. https://doi.org/10.1016/j.matdes.2019.108001
    4. https://doi.org/10.1007/s00170-010-2812-2
    5. https://doi.org/10.1108/RPJ-10-2022-0358
  4. References to Material Extrusion AM techniques
    1. https://doi.org/10.3390/ma10030305
    2. https://doi.org/10.1080/17452759.2024.2331206
    3. https://doi.org/10.1016/j.matlet.2005.04.027
    4. https://doi.org/10.1016/j.addma.2020.101778
    5. https://doi.org/10.18297/etd/3927
  5. References for Material Jetting:
    1. https://doi.org/10.1016/j.addma.2023.103640
    2. 10.1021/acsaenm.4c00796
    3. https://doi.org/10.1016/j.apmt.2025.102938
  6. References to modelling and monitoring in Binder-based AM techniques:
    1. https://doi.org/10.1080/17452759.2024.2331206
    2. https://doi.org/10.1016/j.jmapro.2025.04.076
  7. The manuscript would benefit from a clearer statement of what is novel in this review compared to existing reviews on in-situ monitoring or AE in AM. In the introduction, briefly summarize any related reviews and highlight how this paper provides new insights to justify its contribution.

  8. In several sections, the manuscript tends to catalogue findings from many studies without enough critical synthesis. For instance, Section 3.2 enumerates various correlations between AE signal features and phenomena (spatter, melt pool stability, porosity, cracking, etc.), and Section 3.3 lists numerous machine learning algorithms and their reported accuracies. While the thoroughness is commendable, the review would be stronger if the authors provided more interpretation and comparison. 

  9. After summarizing a group of studies, it is recommended to add a few sentences discussing what general lessons or trends emerging from these studies. For example, are certain AE features consistently identified as the most sensitive to defects? Are there conflicting results between studies that need reconciliation?

Author Response

Response to Reviewer 3 Comments

Comments 1: The authors are encouraged to update the manuscript with state-of-the-art literature in this filed. Especially since this is a literature review paper, it is quentessential for the authors to cite more recent papers in this field of research. Only 2 papers from 2025 and 2 from 2024 are cited. A few publications pertinent to this research are provided in the comments to follow.

Response 1: Thank you for pointing this out. We agree with this comment. Therefore, we have made the necessary revisions.

“The AE online monitoring technology analyzes the sound wave signals generated during the material processing to provide an efficient monitoring method for the AM process. This technology does not require integrating sensors into the processing head, and it is flexible in deployment, enabling low-cost and real-time online process monitoring [50]. In practical applications, Raeker et al. [51] utilized piezoelectric transduce (PZT) sensors and AE technology to achieve in-situ crack detection in a single-channel laser melting experiment. By analyzing the acoustic emission signals, they revealed the influence patterns of different laser powers and scanning speeds on crack size, as well as the crack initiation characteristics of the material. Ansari et al. [52] proposed a monitoring method based on the exponential decay characteristics of AE signals. Through the analysis of the second derivative, noise was effectively filtered out, enabling the identification, quantification of surface and internal cracks during DED, as well as the reliable correlation of their initiation time and location. Furthermore, Xu et al. [53] combined AE technology with machine learning algorithms to systematically analyze the influence of key process parameters such as overlap rate, dwell time, and layer number on the relationship between AE features and Rockwell hardness. By integrating strain sensors, they achieved real-time and non-destructive prediction of the hardness of L-DED components, demonstrating a promising application in engineering.” (Please see to Section 1.2, Paragraph 7. These contents have been modified using blue font.)

Comments 2: While classifying metal additive manufacturing technologies, the authors missed out an another important classification - "Binder-based AM techniques". This class of MAM technologies encompass binder jetting, material extrusion (MEX), and material jetting. Please expand on these technologies in your write-up. A few references to these technologies are provided below.

Response 2: Thank you for pointing this out. We agree with this comment. Therefore, we have made the necessary revisions.

“In addition to the aforementioned AM methods, there are Binder Jetting Additive Manufacturing (BJ-AM) [9], Material Jetting Additive Manufacturing (MJ-AM) [10], and Material Extrusion Additive Manufacturing (MEX-AM) [11]. MEX-AM is widely popular due to its low cost and ease of use; MJ-AM can manufacture high-precision parts with smooth surfaces and multi-material properties through a precise curing process similar to inkjet printing; and BJ-AM uses binders to selectively bond powders, achieving unsupported, full-color, and efficient large-scale manufacturing. These technologies, from different aspects such as economy, precision, and efficiency, jointly support a wide range of industrial and prototyping manufacturing needs.” (Please see to Section 1.1, Paragraph 3. These contents have been modified using purple font.)

Comments 3: References to Binder jetting:

(1)    https://doi.org/10.1108/RPJ-12-2014-0180

(2)    https://doi.org/10.1016/j.promfg.2017.07.084

(3)    https://doi.org/10.1016/j.matdes.2019.108001

(4)    https://doi.org/10.1007/s00170-010-2812-2

(5)    https://doi.org/10.1108/RPJ-10-2022-0358

Response 3: Thank you for pointing this out. We agree with this comment. Therefore, we have made the necessary revisions.

In addition to the aforementioned AM methods, there are Binder Jetting Additive Manufacturing (BJ-AM) [9], Material Jetting Additive Manufacturing (MJ-AM) [10], and Material Extrusion Additive Manufacturing (MEX-AM) [11]. (Please see to Section 1.1, Paragraph 3, lines 1 to 3 These contents have been modified using purple font.)

In the research of BJ-AM, Kumar et al. [12, 13] improved the part density from 92% after sintering to 99.7% of the theoretical density through post-treatment with hot isostatic pressing (HIP), and further prepared copper parts with porosity ranging from 2.7% to 16.4% by adjusting the powder morphology, sintering process, and HIP parameters. They systematically revealed the influence laws of porosity on material properties. (Please see to Section 1.1, Paragraph 4, lines 1 to 6. These contents have been modified using purple font.)

Comments 4: References to Material Extrusion AM techniques

(1)    https://doi.org/10.3390/ma10030305

(2)    https://doi.org/10.1080/17452759.2024.2331206

(3)    https://doi.org/10.1016/j.matlet.2005.04.027

(4)    https://doi.org/10.1016/j.addma.2020.101778

(5)    https://doi.org/10.18297/etd/3927

Response 4: Thank you for pointing this out. We agree with this comment. Therefore, we have made the necessary revisions.

“In addition to the aforementioned AM methods, there are Binder Jetting Additive Manufacturing (BJ-AM) [9], Material Jetting Additive Manufacturing (MJ-AM) [10], and Material Extrusion Additive Manufacturing (MEX-AM) [11].” (Please see to Section 1.1, Paragraph 3, lines 1 to 3. These contents have been modified using purple font.)

“In the field of MEX-AM, Ren et al. [15] systematically studied the key performance parameters such as the rheological behavior of raw materials, the strength of green bodies, and the hardness of sintering through the melt extrusion printing technology, and optimized the printing and sintering parameters using the orthogonal design method.” (Please see to Section 1.1, Paragraph 4, lines 12 to 16. These contents have been modified using purple font.)

Comments 5: References for Material Jetting:

  • https://doi.org/10.1016/j.addma.2023.103640
  • 1021/acsaenm.4c00796
  • https://doi.org/10.1016/j.apmt.2025.102938

Response 5: Thank you for pointing this out. We agree with this comment. Therefore, we have made the necessary revisions.

“In MJ-AM, non-selective ultraviolet (UV) irradiation can easily lead to uneven dose received by components at different heights, causing fluctuations in mechanical properties and batch-to-batch differences. To control the process fluctuations, Bezek et al. [14] developed a process model that combines irradiance distribution and molding configuration. Based on the Bayesian-Lambert theorem, they quantitatively predicted the cumulative UV dose of each printed voxel and established the correlation between dose and mechanical properties through experiments.” (Please see to Section 1.1, Paragraph 4, lines 6 to 12. These contents have been modified using purple font.)

Comments 6: References to modelling and monitoring in Binder-based AM techniques:

  • https://doi.org/10.1080/17452759.2024.2331206
  • https://doi.org/10.1016/j.jmapro.2025.04.076

Response 6: Thank you for pointing this out. We agree with this comment. Therefore, we have made the necessary revisions.

“In terms of process mechanism research, Ajjarapu et al.[16] constructed a process diagram to reveal the important role of back pressure and reflux in the pressure drop of the nozzle, and by means of variance analysis and regression models, clarified the influence of printing conditions on performance and prediction ability.” (Please see to Section 1.1, Paragraph 4, last 4 lines. These contents have been modified using purple font.)

Comments 7: The manuscript would benefit from a clearer statement of what is novel in this review compared to existing reviews on in-situ monitoring or AE in AM. In the introduction, briefly summarize any related reviews and highlight how this paper provides new insights to justify its contribution.

Response 7: Thank you for pointing this out. We agree with this comment. Therefore, we have made the necessary revisions.

“AE monitoring detects elastic waves generated within the material during fabrication, allowing for real-time assessment of the process dynamics. Hossain et al. [55] investigated the progress of acoustic technology in modulation processes and part quality monitoring, and also explored the potential applications of acoustic technology in quality inspection and monitoring of modulation techniques. Compared with the existing reviews, the novelty of this review does not merely lie in the simple listing of acoustic emission technology. Instead, it lies in the first time constructing a deep analysis framework that runs through the entire technical chain of “hardware configuration - parameter correlation - intelligent monitoring - closed-loop control”. It deeply analyzes the unique challenges and solutions faced by acoustic emission technology from signal acquisition, processing to the final realization of intelligent decision-making, filling the gap of the lack of specialized and systematic reviews in this field.” (Please see to Section 1.2, Paragraph 8. These contents have been modified using purple font.)

Comments 8: In several sections, the manuscript tends to catalogue findings from many studies without enough critical synthesis. For instance, Section 3.2 enumerates various correlations between AE signal features and phenomena (spatter, melt pool stability, porosity, cracking, etc.), and Section 3.3 lists numerous machine learning algorithms and their reported accuracies. While the thoroughness is commendable, the review would be stronger if the authors provided more interpretation and comparison.

Response 8: Thank you for pointing this out. We agree with this comment. Therefore, we have made the necessary revisions.

3.2 Section

“During the melting process, the acoustic emission equipment monitors the acoustic signals of the melted and non-melted materials in the time domain and frequency domain. Through time-domain and frequency-domain analysis, noise can be removed and the frequency range of the melting acoustic signals can be determined. At the same time, different agglomeration and rupture phenomena generated under different melting conditions are analyzed [80].” (Please see to Section 3.2.1, Paragraph 6, lines 7 to 13. These contents have been modified using purple font.)

3.3 Section

“The performance variations are attributed to a combination of factors, primarily including the inherent complexity and spectral overlap of AE signatures from certain physical phenomena, which challenge the feature extraction capability of simpler models. Furthermore, the performance of some models was constrained by the limited quantity of high-quality training data available for specific, rare defect types, leading to insufficient feature learning. The suboptimal performance of conventional machine learning models can also be linked to their dependence on hand-crafted features and their limited capacity to capture the complex, non-linear patterns in raw waveform data compared to deep learning architectures.” (Please see to Section 3.3.1.6, Paragraph 3. These contents have been modified using red font.)

Comments 9: After summarizing a group of studies, it is recommended to add a few sentences discussing what general lessons or trends emerging from these studies. For example, are certain AE features consistently identified as the most sensitive to defects? Are there conflicting results between studies that need reconciliation?

Response 9: Thank you for pointing this out. We agree with this comment. Therefore, we have made the necessary revisions.

“Table 1 presents the performance of various modeling algorithms in the online monitoring of AE in MAM, verifying the effectiveness of this technology in ensuring quality. Different algorithms demonstrate significant advantages in specific tasks: Linear regression has an accuracy rate of 98.5% in predicting the powder flow rate of L-DED, demonstrating its efficiency in handling linear problems; K-Means maintains a stable accuracy of 83%–90% in tasks such as defect identification and splashing classification, suitable for unsupervised analysis scenarios; NN achieve an excellent performance of 96.0% in process state recognition, highlighting their ability to handle non-linear features; SVM and RF, although applied to non-core scenarios, also achieve accuracy rates of 91.7% and 97.6% respectively, demonstrating good generalization performance. The results show that by combining the appropriate algorithm strategy, AE monitoring can effectively diagnose key quality issues in the manufacturing process, providing a reliable basis for intelligent control.” (Please see to Section 3.3.1.6, Paragraph 4. These contents have been modified using purple font.)

Reviewer 4 Report

Comments and Suggestions for Authors

Monitoring defect formation during additive manufacturing of metals and alloys using acoustic emission (AE) becomes one of the most promising diagnostic technologies. Although AE is relatively “young” compared to other traditional NDT methods, it has every potential to become an acceptable technology for monitoring defects formed during additive manufacturing of metallic parts and structures. The relevance of this article, as well as the application of AE itself, is beyond doubt. However, for the practical application of AE for this operation, a number of issues must be addressed, most of which are discussed in this review. These include the selection of AE sensor locations and types, and consideration of the heat input source (laser, arc, or electron beam).

There are a number of issues that, in my opinion, should be addressed in more detail in the article.

  1. Although the review mentions electron beam additive manufacturing (PBF-EB), the article does not present sufficient studies describing AE testing using this technology. Electron beam additive manufacturing occupies a special position in this case, since the process occurs in a vacuum, which imposes a number of limitations on the AE monitoring application.
  2. I believe that the ability to assess crack size using the AE method is quite difficult (Section 3.2.2). At least from the data shown in Figures 15 and 16, this opportunity is not obvious. This question needs to be either strengthened or expressed more cautiously.
  3. Section “3.3.2. Closed-Loop Control with Acoustic Emission Monitoring.” The algorithm is unclear: which specific AE parameters are used to monitor the additive manufacturing process. Are these acoustic parameters or process parameters? A more specific description is needed here, perhaps using a specific example.
  4. Figure 8 addresses the important issue of considering piezoelectric sensors in AE monitoring. However, firstly, it is difficult to read this figure. Secondly, it doesn't answer the question of what principles should be used when placing sensors during AE monitoring. Since the acoustic path significantly influences AE monitoring results, this issue needs to be examined in more detail.
  5. This may be a display error, but Figures 5, 7, 8, 10, and 12 are of insufficient quality.
  6. Figure 19, in my opinion, is uninformative. What does it show?

Overall, this is a good review that could use some improvement.

Author Response

Response to Reviewer 4 Comments

Comments 1: Although the review mentions electron beam additive manufacturing (PBF-EB), the article does not present sufficient studies describing AE testing using this technology. Electron beam additive manufacturing occupies a special position in this case, since the process occurs in a vacuum, which imposes a number of limitations on the AE monitoring application.

Response 1: Thank you for pointing this out. We agree with this comment. Therefore, we have made the necessary revisions. The manuscript, when describing the classification of metal additive manufacturing, mentioned electron beam additive manufacturing. Currently, there are relatively few studies on in-situ acoustic emission monitoring of electron beam additive manufacturing. Mainly, internal forming quality is predicted through machine learning. We have made revisions to the manuscript in the corresponding section.

“Kuang et al. [137] employed kernel functions to map the original feature space into a high-dimensional space, thereby enabling nonlinear classification and regression.” (Please see to Section 3.3.1.5, Paragraph 2, Last 3 lines. These contents have been modified using green font.)

Comments 2: I believe that the ability to assess crack size using the AE method is quite difficult (Section 3.2.2). At least from the data shown in Figures 15 and 16, this opportunity is not obvious. This question needs to be either strengthened or expressed more cautiously.

Response 2: Thank you for pointing this out. We agree with this comment. Therefore, we have made the necessary revisions.

“The signal characteristics corresponding to each defect are different. Fig. 15 (a-c) respectively represent the minimum defect (class 1), crack defect (class 2), and porosity (class 3).” (Please see to Section 3.2.2, Paragraph 5, lines 13 to 15. These contents have been modified using blue font.)

“Two crack signals appeared at the 5th and 8th minutes of the manufacturing process. They seemed to correspond to the cracks near the substrate at the cutting site. The first crack (at a height of 4 mm) appeared to be associated with the AE signal recorded at the 18th minute. This indicates that crack defects can be monitored through the time-domain characteristics of acoustic emission signals.” (Please see to Section 3.2.2, Paragraph 6, lines 6 to 11. These contents have been modified using green font.)

Comments 3: Section “3.3.2. Closed-Loop Control with Acoustic Emission Monitoring.” The algorithm is unclear: which specific AE parameters are used to monitor the additive manufacturing process. Are these acoustic parameters or process parameters? A more specific description is needed here, perhaps using a specific example.

Response 3: Thank you for pointing this out. We agree with this comment. Therefore, we have made the necessary revisions.

“Some researchers have already achieved substantial accomplishments in using machine learning based on these acoustic features for defect detection. Chen et al. [141] proposed a convolutional neural network based on Mel-Frequency Cepstral Coefficients (MFCC) features Mel-Frequency Cepstral Coefficients convolutional neural networks (MFCC-CNN) for L-DED sound classification, aiming to detect cracks and keyhole pores. The overall accuracy rate reached 89%. This model outperformed traditional machine learning methods such as SVM and KNN in terms of keyhole pore recognition accuracy (93%) and AUC-ROC (98%). The study also indicated that noise removal of acoustic signals can effectively improve classification performance, which is of great significance for online monitoring and feedback control of acoustic emission.” (Please see to Section 3.3.2, Paragraph 1, lines 9 to 19. These contents have been modified using green font.)

Comments 4: Figure 8 addresses the important issue of considering piezoelectric sensors in AE monitoring. However, firstly, it is difficult to read this figure. Secondly, it doesn't answer the question of what principles should be used when placing sensors during AE monitoring. Since the acoustic path significantly influences AE monitoring results, this issue needs to be examined in more detail.

Response 4: Thank you for pointing this out. We agree with this comment. Therefore, we have made the necessary revisions.

“The installation position of the PZT sensor in the additive manufacturing process is shown in Fig. 8. The principle of the AE PZT sensor is based on the positive piezoelectric effect. It converts the transient elastic stress waves (AE signals) generated due to damage (such as crack propagation, fiber fracture) within the material into electrical signals. The detection object selects a sensor with an appropriate resonant frequency and sensitivity, and ensures excellent acoustic coupling between the sensor and the surface of the component by using coupling agent and constant clamping force. In application, setting reasonable threshold values and parameter analysis is used to distinguish real AE events from noise. At the same time, it must be calibrated with standard AE sources (such as broken lead) to ensure the accuracy of data acquisition and the repeatability of the results.” (Please see to Section 3.1, Paragraph 4. These contents have been modified using green font.)

Comments 5: This may be a display error, but Figures 5, 7, 8, 10, and 12 are of insufficient quality.

Response 5: Thank you for pointing this out. We agree with this comment. Therefore, we have made the necessary revisions.

Fig. 5. Classification of AE sensors.

Fig. 7. Configuration of FBG AE monitoring [67, 68]: (a) Mounted on the sidewall of an SLM build chamber; (b) AE signal acquisition system; (c) Schematic diagram of the sensing principle.

Fig. 8. Mounting locations of PZT AE sensors: (a-c) on the substrate surface in SLM;

(d-e) on the substrate surface in L-DED; (g) beneath the substrate platform in WA-DED.

Fig. 10. Correlation analysis of AE parameters with process variables: (a) RMS value versus powder flow rate in L-DED, where represents the RMS of the AE signal and  denotes the actual powder flow rate; (b) AE signal intensity as a function of wire feed speed for AA5087 and AA6060 aluminum alloys under CMT and CMT+P modes.

Fig. 12. Frequency domain diagrams of molten pool sputtering images and sound signals under different laser powers: (a) low laser power; (b) medium laser power; (c) high laser power.

Comments 6: Figure 19, in my opinion, is uninformative. What does it show?

Response 6: Thank you for pointing this out. We agree with this comment. Therefore, we have made the necessary revisions.

“Song et al. [152], used a dual-color high-temperature meter to measure the molten pool temperature during laser cladding. The dynamic relationship between laser power and the molten pool was described using a state-space model. The temperature was experimentally identified using subspace methods. The closed-loop process can track the molten pool temperature to the reference temperature curve by adjusting the diode to control the laser power. The compensatory deficiency was verified using cladding for GPC. as shown in Fig. 19.” (Please see to Section 3.3.2, Paragraph 3. These contents have been modified using red and blue font.)
